# Examining regional groundwater-surface water dynamics using an integrated hydrologic model of the San Joaquin River basin

James M. Gilbert[1,2*], Reed M. Maxwell[1,2]

[1]Hydrologic Science and Engineering Program, Geology and Geological Engineering Department, Integrated GroundWater Modeling Center, Colorado School of Mines, Golden, 80401, USA
[2]Climate Change Water and Society (CCWAS), Integrative Graduate Education and Research Traineeship (IGERT), USA
*Now at the United States Bureau of Reclamation, Technical Service Center, Denver, 80225, USA

Correspondence to: James M. Gilbert (jmgilbert@usbr.gov)

**Abstract.** Widespread irrigated agriculture and a growing population depend on the complex hydrology of the San Joaquin River basin in California. The challenge of managing this complex hydrology hinges, in part, on understanding and quantifying how processes interact to support the groundwater and surface water systems. Here, we use the integrated hydrologic platform ParFlow-CLM to simulate hourly 1-km gridded hydrology over one year to study unimpacted groundwater-surface water dynamics in the basin. Comparisons of simulated results to observations show the model accurately captures important regional-scale partitioning of water among streamflow, evapotranspiration (ET), snow, and subsurface storage. Analysis of this simulated Central Valley groundwater system reveals the seasonal cycle of recharge and discharge as well as the role of the small but temporally constant portion of groundwater recharge that comes from the mountain block. Considering uncertainty in mountain block hydraulic conductivity, model results suggest this component accounts for 7%-23% of total Central Valley recharge. A simulated surface water budget guides a hydrograph decomposition that quantifies the temporally variable contribution of local runoff, valley rim inflows, storage, and groundwater to streamflow across the Central Valley. Power spectra of hydrograph components suggest interactions with groundwater across the valley act to increase long-term correlation in San Joaquin River outflows. Finally, model results reveal hysteresis in the relationship between basin streamflow and groundwater contributions to flow. Using hourly model results, we interpret the hysteretic cycle to be a result of daily-scale fluctuations from precipitation and ET superimposed on seasonal and basin-scale recharge and discharge.

## 1 Introduction

A region's water resources reflect the unique combination of climate, geology, ecology, and human activity particular to that place, the emergent result of the cumulative and universal biogeochemical processes occurring therein. With rising (and competing) priorities for allocation of water resources for human and environmental needs, examination of these processes across watersheds (catchments, basins) and how they transform precipitation into much relied-upon surface and groundwater

supplies is increasingly salient. In light of this, we pursue insights into regional watershed function by applying an integrated hydrologic model to the San Joaquin River basin in California.

The San Joaquin River basin is a fitting subject of study, as it typifies issues common to many water resource systems not only in the American West but many mountain-valley agricultural systems worldwide. Runoff from mountain snowpack and

extraction of groundwater from the Central Valley aquifer system has facilitated a massive transformation of the San Joaquin system from its state prior to European settlement—much of the Valley area has been converted from native vegetation to agricultural and urban uses. The basin now supports more than $20 billion in annual agricultural production (California Department of Food and Agriculture (CDFA), 2015) while hosting a population that is projected to urbanize and grow 67-143% from 2006 levels by 2050 (California Department of Water Resources, 2014). The legacy of past water uses and

growth of future demands, coupled with declining and variable snowpack and over-extraction of groundwater, thus pose escalating challenges to water management in the region.

These challenges have motivated much previous research in the San Joaquin basin. Early studies, motivated by burgeoning irrigation and agricultural production objectives in the Central Valley, provide coarse estimates of surface and groundwater supplies while hinting at system function during early stages of regional development (Alexander et al., 1874; Hall, 1886a,

1886b; Harding and Robertson, 1912; Jones, 1934; McGlashan, 1930; Mendenhall et al., 1916). Recognition of the importance of, and growing impacts of development on, the Central Valley aquifer system prompted more extensive and detailed data collection and study throughout the Central Valley (Bertoldi et al., 1991; Davis et al., 1959, 1964; Mullen and Nady, 1985; Page, 1986), eventually leading to regional groundwater modeling studies. Williamson et al. (1989) constructed the first regional groundwater model of the Central Valley aquifer and presented complete pre- and post-development water

budget estimates for the system. Subsequent modeling efforts have extended from this pioneering effort: Faunt (2009) and Hanson et al. (2010, 2012) use MODFLOW and the Farm Process (FMP) module while the California Department of Water Resources (CA-DWR) has developed the C2VSim (Brush et al., 2013; Dale et al., 2013) model (based on the Integrated Water Flow Model framework) to simulate the coupled evolution of the Central Valley aquifer system under irrigation development. Although implementation differs in each, these models share similar spatial extents (i.e. limited to the Central

Valley), are designed toward multi-decadal simulation of agricultural budgets and associated hydrologic impacts, and have relatively coarse temporal discretization. While this approach provides a useful basis for understanding trends in the groundwater system, it affords less insight into the interactions among component of the underlying natural hydrologic system. The steady-state integrated hydrologic model of Bolger et al., (2011) provides a general picture of such a pre-development system but cannot be used to assess temporal or mountain-valley dynamics.

The dependence of the Central Valley on Sierra Nevada precipitation, seasonal snowpack, and runoff has motivated numerous studies examining aspects of San Joaquin River basin hydrology. To date, most simulation studies investigate projected climate change impacts to snowpack, ET, and streamflow over broad scales (e.g., Das et al., 2011; Dettinger et al., 2004; Ficklin et al., 2009, 2012; Godsey et al., 2014; Lettenmaier and Gan, 1990; Null et al., 2010; VanRheenen et al., 2004). With the focus of these macro-scale studies aimed at preserving an ensemble of projected climate conditions or

propagating effects to the managed water infrastructure of the Central Valley, the results provide more limited insight into the hydrologic mechanisms at work over finer spatial and temporal scales in the system. The 'Basin Characterization Model' (BCM) (Flint et al., 2013) improves spatial resolution of unimpaired hydrology across the entire Central Valley drainage, but its monthly temporal resolution make it better suited for longer-period temporal variations.

Despite the numerous studies of the region, there remains a significant gap in the understanding of how physical hydrologic processes interact over short spatial and temporal scales to yield variation in important water resource components. Therefore, the objective of this work is to study the mechanisms that interact to affect streamflow and groundwater in the San Joaquin River basin over hydrologically relevant timescales of hours to months. We do so using the San Joaquin Basin Model (SJBM), constructed using the ParFlow-CLM integrated hydrologic simulation platform (Ashby and Falgout, 1996;

Jones and Woodward, 2001; Kollet and Maxwell, 2006, 2008; Maxwell, 2013; Maxwell and Miller, 2005). The model serves as a means to analyse specific interactions that are difficult to study with measurements alone: mountain block recharge to the Central Valley, temporal dynamics of the connected subsurface-surface system and its component fluxes and stores, and groundwater-surface water exchange over a basin scale. The following sections describe our conceptual model of the San Joaquin River basin, an overview of model construction, a comparison of simulated results to observational data, and

analysis of phenomena revealed by the simulation.

## 2 Conceptual model

The San Joaquin basin covers approximately 40,400 $km^2$ and encompasses hydrologic regimes ranging from the relatively moist, snow-dominated montane system of the Sierra Nevada Mountains to the semi-arid to arid Central Valley and eastern Coast Ranges (Fig. 1). We conceive the San Joaquin River basin hydrologic system as a system of interconnected surface

and groundwater components that respond differentially to landscape factors to store and transmit water, with the residual of primary incoming and outgoing fluxes realized as streamflow or changes in groundwater levels. This conceptualization views the apparent differences in hydrology within the Sierra Nevada mountains and the Central Valley regions as an emergent property of the same fundamental processes across the landscape rather than evidence for different processes in different regions. Therefore we assume the same processes are active everywhere, but local properties (e.g. meteorology,

slope, vegetation, subsurface) determine which ones dominate at a particular location and time.

Precipitation is the predominant inflow to the San Joaquin River basin, with most of it concentrated on the western face of the Sierra Nevada Mountains. Within the San Joaquin River drainage, average annual precipitation on the Central Valley floor and Coast Ranges varies from 20-30 cm while the Sierra Nevada receives 90 cm to more than 150 cm at high elevations (Brush et al., 2013; PRISM Climate Group and Oregon State University, 2015). Much of the mountain

precipitation is stored as a seasonally accumulating snowpack on the land surface while the smaller portion that falls within the valley supports soil moisture, recharge, ET, and streamflow. The dominant outflows from the system are evapotranspiration (ET) (approximately 125-145 cm/yr on the Valley floor, (Faunt, 2009)) and streamflow (estimated to be

about 11 cm/yr at the Vernalis gage, https://waterdata.usgs.gov/ca/nwis/inventory/?site_no=11303500&agency_cd=USGS).

Some groundwater likely leaves the basin, although the amount is difficult to quantify along basin boundaries in the Sierra Nevada and in the Central Valley for predevelopment hydrologic conditions. We consider groundwater outflow to be a minor outflow component given that the approximated predevelopment water table of (Williamson et al., 1989) shows a

relatively flat water table in the region coinciding with the southern basin in the Central Valley and that Faunt (2009) shows that groundwater flow to the Delta (leaving the basin to the north) as a fraction of the overall basin groundwater budget is comparatively small. The mountain headwaters of the San Joaquin River and its main tributaries (the Stanislaus, Tuolumne, & Merced Rivers) arise near the crest of the Sierra Nevada Range from a mix of spring snowmelt, seasonal storm events, and baseflow. The dry east face of the Coastal Ranges contributes comparatively minimal streamflow through intermittent

streams (Mullen and Nady, 1985; Nady and Larragueta, 1983). The San Joaquin River and its east-side tributaries exit the mountains and flow across a deep assemblage of heterogeneous unconsolidated sediments deposited in the near surface as coalescing alluvial fans, floodplains, alluvium, and lacustrine silts and clays of variable extent and depth (Burow et al., 2004; Farrar and Bertoldi, 1988; Laudon and Belitz, 1991; Page, 1986; Williamson et al., 1989). The Corcoran Clay is the most laterally extensive of these clay deposits and has been mapped across much of the western San Joaquin valley (Davis et al.,

1959; Page, 1986; Williamson et al., 1989). In this study we conceptualize the Central Valley sediments as comprising a laterally heterogeneous but continuous aquifer system (Faunt, 2009; Williamson et al., 1989).

The Coast Range and Sierra Nevada mountain blocks cover a large portion of the basin and are considered in this study to be hydraulically-connected parts of the basin system with spatially-variable properties related to their distinct geologies. The complex lithologies of the Coast Range (dominated by marine sedimentary and metasedimentary rocks) and Sierra

(predominantly granite and other plutonic rocks) (Farrar and Bertoldi, 1988; Jennings, C.W. et al., 1977), are assumed to have some non-zero permeability through a depth of 500 meters, and that, where this permeability is the result of fractures, the hydraulic properties of these fractures can be adequately represented by a block effective parameter. This conceptualization of mountain block flow systems as a fractured (or otherwise permeable) mantle overlying relatively impermeable crystalline rock is consistent with recent approaches to hydrologic simulation in such systems (Manning and

Solomon, 2005; Welch and Allen, 2014). We assume the mountain block transmits water internally through local and regional topographically-driven subsurface flow paths and is connected to the aquifers of the Central Valley through mountain front recharge and diffuse mountain block recharge (Gleeson and Manning, 2008; Wilson and Guan, 2013).

We conceptualize the system in its quasi-predevelopment state, unimpaired by the influence of groundwater pumping, stream impoundments and reservoirs, or surface water diversions. The rationale for considering a predevelopment condition

is two-fold: first, the paucity of data that predate widespread hydrologic change limits the ability to reconstruct the underlying natural condition of the system; and second, understanding the extent of change embedded in the modern San Joaquin River basin hydrology requires a baseline un-impacted condition with which to compare. Under our conceptualized predevelopment conditions, the Central Valley riparian environment is characterized by seasonal flooding across wide flat floodplains and a near-surface water table that maintains perennial stream flow and wetlands after snow melt flows have

subsided in late summer and fall. Recharge to the Valley groundwater system is assumed to occur by overbank and channel recharge from high snowmelt flows throughout the valley, local mountain front recharge as rivers discharge from granite dominated to sediment-dominated subsurface at the valley wall, as well as diffuse recharge from valley precipitation events and mountain block recharge. This recharge and the general northwest-trending topographic gradient drive regional groundwater flow toward the San Joaquin River mainstem and the Sacramento-San Joaquin Delta.

## 3 Model construction

The construction and spin-up of the SJBM is summarized in the following paragraphs. A more detailed description of the inputs and processing used to develop the model is provided as Appendices A through D.

### 3.1 ParFlow-CLM simulation framework

We use the ParFlow-CLM (PF.CLM) code (Ashby and Falgout, 1996; Jones and Woodward, 2001; Kollet and Maxwell, 2006; Maxwell, 2013; Maxwell and Miller, 2005) to simulate variably saturated subsurface flow, overland flow, and land surface processes (evaporation, transpiration, & snowpack dynamics) for the San Joaquin River basin. The ParFlow (PF) portion of the code implicitly and simultaneously solves the governing equations for subsurface and overland flow systems: the Richards' equation for variably saturated subsurface flow and the kinematic approximation of the shallow water equations at the land surface. No river parameterization is used—channel and overland flow are simulated according to a uniform application of the kinematic approximation in each surface cell in the domain. The CLM component, a version of the Common Land Model (Dai et al., 2003) modified for use with ParFlow (Maxwell and Miller, 2005), simulates water and energy fluxes at the land surface and is coupled via pressure and saturation in the top model layers of the ParFlow model grid. Coupled PF.CLM simulations are commonly run at one-hour time steps; this convention is used in this study as well.

### 3.2 Summary of SJBM construction

The SJBM is defined on a 1-km lateral resolution grid with 270 columns and 220 rows and five variable-thickness layers covering the top 500 m from the land surface. The vertical discretization is refined over the top four layers (thicknesses are 0.1 m, 0.3 m, 0.6 m, 1.0 m, respectively) and simplified in the bottom (498 m thick) layer. This configuration aggregates vertical heterogeneity in the Central Valley aquifer system while maintaining large-scale lateral heterogeneity, resulting in a model that reflects the lateral component of regional flow but that does not resolve changes in vertical gradients at depth. Boundary conditions are no-flow on the lateral and bottom faces and an overland flow condition at the land surface. Inputs include permeability, van Genuchten constitutive relationship parameters, porosity, and specific storage for the subsurface and topographic slopes, Manning's roughness values, and vegetation type and properties at the land surface. Subsurface properties are assigned based on hydrostratigraphic indicator categories aggregated from previous studies (Faunt, 2009; Mansoor, 2009) with corresponding hydraulic property values assigned according to studies (Faunt, 2009) where available

and from reference or literature values elsewhere. At the surface, slopes are derived from a digital elevation model, vegetation is mapped from remote sensing products, and Manning's roughness values are assigned based on reference values.

### 3.3 Spin-up and simulation

We initialize the SJBM through a "spin-up" process in which the simulated hydrology is brought to dynamic equilibrium with a meteorological forcing through iterative simulations. We use a two-stage spin-up process that first applies a constant-in-time net precipitation flux to generate an approximate regional flow system followed by iterative one-year simulations using hourly meteorological forcing. We define water year 2009 (October 1, 2008 to September 30, 2009, WY2009) as our period of interest for simulation and thus use hourly meteorological forcing for this year to drive the spin-up runs and the

subsequent analysis run on which the following observation comparisons and analyses are based. Water year 2009 was chosen because it was both approximately average (in terms of precipitation and temperature) and was recent enough that a range of *in situ*, remotely-sensed, and related data products provide measurements and coverage of a range of basin hydrologic components. While a perfectly "average" climatological year is rare, water year 2009 was relatively close: average annual temperature was slightly warmer (+1°C) and annual precipitation was slightly drier (-9.9 cm) than the 1895-

1970 historical averages (NOAA National Centers for Environmental Information, 2016).

### 4. Comparison to observations

We build confidence in the SJBM by comparing simulated hydrology to observational equivalents derived from the real-world San Joaquin River system. Here we utilize available measurements and data products from stations and remote sensing systems to provide estimates of the key components of the system water budget being simulated. Specifically, we compare

simulated hydrology to: 1) estimates of predevelopment water table elevation; 2) estimates of unimpaired runoff ratios at gage sites; 3) station and data assimilation-derived snow pack properties; 4) remotely sensed evapotranspiration (ET); and 5) remotely sensed amplitude of terrestrial water storage variation. Given that the historic measurement record coincides with widespread modification of the hydrologic system, observations representative of an undeveloped state are relatively rare. Modern data that capture the combined effects of natural variation and local development provide better temporal and spatial

coverage but are not strictly comparable to the results of the unperturbed simulated system. Nonetheless, considering aggregate behavior at a regional scale (1000-10000 km$^2$) that includes the less-impacted Sierra Nevada reduces some of the overall effect of local hydrologic perturbations in the Central Valley sufficient to permit reasonable comparison of simulated and observed variables, in particular their variation over a full annual cycle. These comparisons, caveats notwithstanding, help indicate the high degree of consistency by which the ParFlow-CLM model captures important underlying processes in a

real, complex system.

### 4.1 Predevelopment water table

Williamson et al. (1989) mapped a predevelopment water table in the Central Valley based on water depth and elevation measurements recorded in the late 1800s and early 1900s. We compare this predevelopment water table map to the average annual water table elevation as simulated in the lowest model layer of the SJBM in Fig. 2. The simulated and mapped water tables are generally consistent across much of the Central Valley, with the SJBM matching the large scale features of the eastern valley as shown by (Williamson et al., 1989). In particular, the SJBM matches the higher water table elevations along the southeast valley edge that grade toward the valley axis locally and toward the Sacramento-San Joaquin delta (to the northeast) regionally. The comparison of water table elevations also highlights several local discrepancies. First, the simulated water table elevations are lower along the western boundary of the Central Valley when compared to the water table contours from Williamson et al (1989). We attribute this difference to two factors: 1) a dry bias in the atmospheric forcing used to drive the model, a bias that yields minimal potential recharge along the already comparatively dry western portion of the Central Valley and 2) an offset introduced to the simulated water table elevations through the use of a modern digital elevation model that incorporates lower land surface elevations compared with those of a predevelopment condition without widespread western Valley subsidence (Riley and Galloway, 1999).

Another difference in the water table elevation contours is apparent along portions of the San Joaquin and several of its tributaries. The SJBM simulates water table configurations characterized by upstream pointing 'v's, indicating groundwater contributions to the channel in those locations. Such deflections of water table contours are absent in the Williamson et al (1989) dataset. This suggests either a localized mismatch in the simulated groundwater-channel head gradient, possibly owing to the one-kilometer grid resolution used, or a failure in the observed water table map to resolve the fine-scale spatial variation in near-channel water table elevations from sparse point measurements. The general qualitative alignment of gaining and losing reaches with the few available (and human-impacted) measurements (summarized in Faunt, 2009, p.170–171) however provides some confidence that the SJBM is simulating broad-scale stream-aquifer interactions acceptably well on the Central Valley floor.

### 4.2 Runoff and streamflow

Rivers in the modern San Joaquin basin have been extensively modified from their natural state. Discharge records, especially at gages in the Central Valley and foothills, largely reflect impacts from a range of water use and water management activities, including: reservoir retention and releases, canal diversions and return flows, and depletion of streamflow by groundwater extraction. Given that we simulate the river basin without these modifications, few datasets exist that provide an appropriate comparison to the model. Two options include: 1) streamflow measurements that predate much of the historic hydrologic modification and 2) modern streamflow measurements that are adjusted to account for the diversion and retention of flow. With regards to the first option, the records compiled by Hall (1886a) provide monthly estimates of flow on several streams and rivers discharging from the Sierra Nevada mountains in the San Joaquin basin prior

to large scale reservoir construction and groundwater extraction (WY 1879-1884). Data to support the second option are available via the California Department of Water Resources' (CA-DWR) estimates of unimpaired or "full natural flow" (FNF). Monthly FNF data are calculated for select rivers by adjusting for gains and losses associated with retention and diversion, are available for most of the 20th and 21st century (length of record depends on the station), and reflect the contemporary climatology (California Data Exchange Center, 2015).

These estimates of unimpacted streamflow have limitations for comparison to the SJBM results. The historic streamflow values are based on measurements of unknown provenance (i.e. the method and frequency of measurements used to develop monthly mean streamflow values are not reported). Furthermore, these data are inherently inconsistent with the model results because they are the result of a historical climatology rather than the modern atmospheric conditions used to force the SJBM. The calculation of FNF data depends heavily on the availability of data for flow corrections. Infrequent or, in the case of groundwater depletions to streamflow, unavailable data mean the FNF datasets cannot represent natural conditions at all locations and all times. Despite these limitations, the Hall (1886a) and FNF records can be useful surrogates for predevelopment flow conditions in that they describe the bounds of expected flow variability. Comparisons between the unimpaired observational record and the flow simulated by the SJBM at corresponding gage locations shows the model tends to under-predict monthly flow volumes (not shown).

However, comparisons of absolute flow rates or volumes can be affected by inconsistencies between model forcing precipitation and the real-world precipitation that produced the observed runoff. To minimize such inconsistencies, we computed the runoff ratio, i.e. streamflow volume as a fraction of monthly precipitation, for 12 CA-DWR FNF stations within the model domain for which approximately 100 years of flow records are available. We tabulated monthly precipitation volume within each watershed upstream of the 12 FNF stations using the resampled monthly PRISM precipitation product for years 1910-2010 (PRISM Climate Group and Oregon State University, 2015). The observed runoff ratio was calculated as the fraction of the monthly precipitation that was realized as unimpaired flow at the FNF station. Similarly, we calculated the simulated runoff ratio for each month within the water year 2009 simulation period using the NLDAS precipitation field and simulated runoff from the SJBM at the 12 corresponding FNF stations. Comparisons between runoff ratios calculated from the historical record and the SJBM are shown in Fig. 3. The dashed line indicates the historical record for WY2009 if available. Note that because the historical contribution of snowmelt to runoff is not precisely known, the calculated runoff ratio was not adjusted to account for snowpack accumulation or melting. In other words, the amount of precipitation received in a month was used to calculate the runoff index for that month. A key consequence of this approach is that the runoff ratio tends to be low during snowpack accumulation months (December – April) but can then be quite high during subsequent months as late-spring and early summer streamflow is generated by snowmelt rather than precipitation. For this reason, the plots in Fig. 3 are shown with logarithmic vertical axes. The plots show that the SJBM reproduces the seasonal variability of runoff ratio and, with the exception of the Stanislaus River, produces runoff ratio magnitudes consistent with the range of historic monthly variability. The match between simulated and historic runoff ratios provides evidence that the SJBM adequately represents the aggregate precipitation-snowpack-runoff mechanisms at a monthly and

watershed scale while suggesting the simulated shortfall in *absolute* streamflow volume may be the result of an issue unrelated to model physics, such as a dry bias in the meteorological forcing used. This precipitation bias is discussed in more detail in Section 4.3 in the context of snowpack.

### 4.3 Snowpack

Comparisons of simulated to observed snowpack properties like snow covered area and snow water equivalent (SWE) allow assessment of the model's ability to capture the energy and hydrologic components of the snow-dominated Sierra mountain system within the SJBM model domain. The observational record of such snow properties includes individual snow station measurements and inferred snowpack characteristics derived from remotely sensed data. In Fig. 4, we compare the SJBM simulated snowpack to the spatial (A) and temporal estimates of SWE (B) and snow covered area (C) provided by the

SNODAS data product (National Operational Hydrologic Remote Sensing Center, 2004) and point measurements of SWE at 33 snow stations (D), available from the California Data Exchange Center (http://cdec.water.ca.gov).

In general, simulated snowpack properties compare favorably with available observations. The temporal patterns in simulated SWE (Fig. 4-B) track closely with the observations while both temporal patterns and magnitude of snow covered area (Fig. 4-C) match the observations very well. The spatial pattern of snow accumulation is, in general, consistent with the

observations although the spatial comparison between simulated and SNODAS April 1 SWE in Fig. 4-A shows that the SJBM is generally biased toward underprediction of SWE accumulation over most of the Sierras. This low SWE bias is apparent in the assimilated SWE data product time series and the station data as well. Fig. 4-B shows that, with the exception of early season snow events, the SJBM simulates a lower domain-total SWE volume than represented by the SNODAS estimate. Fig. 4-D shows under-prediction of SWE at station locations across the Sierra Nevada Mountains. In particular, the

station data show a trend toward increasing SWE discrepancy with increasing SWE depth. The consistency between simulated and observed snow covered area suggests the energy component of the PF.CLM snowpack processes are being represented properly. In contrast, the low SWE bias indicates a shortcoming in the NLDAS precipitation product applied to the complex terrain of the Sierra Nevada Mountains. This phenomenon is not unique to the SJBM – precipitation products gridded at a coarser resolution than terrain and based on sparse station data are prone to under-resolve orographic effects

(Pan, 2003) and are cited as important sources of uncertainty for hydrologic simulation across the western US in general (Mo et al., 2012; Xia et al., 2016) and the Sierra Nevada specifically (Guan et al., 2013; Lundquist et al., 2015).

### 4.4 Evapotranspiration

Evapotranspiration (ET) is the largest outgoing flux over the SJBM domain. Estimates of this flux can be constrained using remotely sensed data-driven estimates like the 8-day 1-$km^2$ MOD16 global ET product (Mu et al., 2007, 2011, 2015). We

mapped each 8-day ET total from the MOD16 product (available via http://www.ntsg.umt.edu/project/mod16) to the SJBM grid and aggregated simulated total ET on the corresponding 8-day intervals. Comparisons between the simulated and observed ET over the SJBM domain are shown in Fig. 5. The SJBM matches the seasonal pattern of domain-aggregated ET

very well (Fig. 5-B) with differences arising only during short time-scale variations. Despite the good temporal match, spatial differences in annual cumulative ET (Fig. 5-A) highlight zones of mismatch between the model and the ET product. First, ET simulated along main river channels is high relative to the MODIS product. This difference can be explained as a combined result of 1) higher simulated ET due to the increased and sustained flow simulated under predevelopment conditions in the model, 2) higher simulated ET as a result of river channels being represented as one kilometre wide in the model, and 3) an inability of the MOD16 algorithm to fully resolve fine-spatial scale ET variation over a given eight day period. Figure 5-A also shows the simulated ET is generally low outside of the riparian corridor within the Central Valley. One explanation for this is that the remotely sensed data on which the MOD16 ET product is based was collected under irrigated conditions, leading to wetter soils and higher ET fluxes, especially in the summer months, relative to the more natural soil moisture distribution simulated in the SJBM. Taken in aggregate, however, the spatial differences between the MOD16 ET product and simulated ET seem to balance as domain total ET volumes track each other closely over water year 2009 (Fig. 5-B). The histogram of 8-day ET values for each cell in Fig. 5-C shows that this apparent consistency between simulated and observed ET volumes at the domain-scale is the result of offsetting differences in the range of ET variability: the SJBM includes more zones that more frequently produce high (>0.03 m) or low (<0.005 m) 8-day ET flux while the MOD16 product is characterized by higher frequencies of mid-range 8-day ET fluxes (0.005-0.015 m). Overall, this comparison suggests that the ET component of the aggregate water budget is simulated within reasonable bounds while underscoring the potential difficulty in parsing the MOD16 ET product for accurate ET comparisons over smaller spatial and temporal scales.

### 4.5 Terrestrial water storage

Terrestrial water storage changes reflect differences in primary input (precipitation) and output (ET) fluxes. While water level records from monitoring wells can be used to estimate subsurface storage changes, such measurements are often too scattered to support regional generalization and, especially in the Central Valley, are impacted by groundwater extraction and the redistribution of water via irrigation. These issues preclude the direct comparison of subsurface storage as estimated from modern monitoring wells with the predevelopment conditions simulated in the SJBM. Furthermore, changes in storage estimated from groundwater wells do not account for changes in other large terrestrial water stores – like lakes, channels, soil moisture, and snow – which may be characterized by large seasonal changes. As an alternative, the terrestrial water storage (TWS) change estimates derived from Gravity Recovery and Climate Experiment (GRACE) satellites (Landerer and Swenson, 2012; Swenson, 2012; Swenson and Wahr, 2006) provide an integrated measure of storage change over the SJBM domain. The GRACE TWS product is not without its limitations for comparison to the SJBM: the GRACE product has a relatively coarse spatial resolution and is based on the contemporary managed hydrologic system rather than the predevelopment conditions simulated. However, as we are interested in the ability of the SJBM to simulate the seasonal variation of TWS rather than determining a precise volume of water stores, the GRACE product provides a valid benchmark for comparison.

The comparison was performed following a sequence of operations. We downloaded the three estimates (CSR, JPL, and GFZ) of the RL05 version of monthly gridded TWS anomaly files for water year 2009 from the GRACE data portal (http://grace.jpl.nasa.gov/data/get-data/monthly-mass-grids-land/). We scaled and resampled the monthly TWS anomalies to the SJBM model domain and calculated the combined measurement and leakage error for the region following the methods

described in Landerer and Swenson (2012). We calculated the mean of the three TWS estimates and applied the error to that average. The TWS anomaly for each of the three estimates and their mean was rescaled to be zero at beginning of water year 2009 for comparison with the model (this is equivalent to removing the time-mean for water year 2009 and shifting by the difference between the mean and the value for October 2008).

Figure 6 shows the comparison between the GRACE TWS anomalies and the corresponding water storage anomalies as

simulated by the SJBM, separated into subsurface, surface, and snow water storage components. Due to the relatively small model footprint and the discrepancies in resolution between the model and the GRACE data, a comparison of absolute storage change for a given month is not supported. Rather, the seasonal amplitude of storage change provides a useful metric for comparison as this represents the larger spatial and temporal scale hydrologic dynamics across the region. For this analysis, the average annual amplitude of total terrestrial water storage derived from GRACE is estimated to be 18.75 cm,

roughly 2.5 cm higher than the 16.27 cm storage change amplitude simulated by the SJBM. With the average error of the GRACE TWS product calculated to be 3.09 cm over the model domain, the difference in amplitudes may be explained by the product uncertainty. The SWE and streamflow comparisons described above suggest a shortfall in precipitation input to the model. Considering that SWE is a large component of the total terrestrial water storage anomaly from the SJBM, this shortfall (roughly 1.7 cm when comparing peak simulated and SNODAS SWE, Fig. 4) may provide a better explanation of

the annual amplitude difference.

The impacts of groundwater extraction, irrigation, and surface water retention in reservoirs across the San Joaquin Basin are embedded in the GRACE signal but not included in the SJBM simulations. For this reason a perfect match between the model and the GRACE product would not be expected, but the amount by which the two *should* differ is considerably uncertain. Uncertainty exists because quantifying this difference requires disentangling the effects of surface water imports

(and exports) to (from) the basin, and consumptive use, seepage, and recharge losses associated with local and imported irrigation water from the underlying natural system. Paired models that accommodate both natural and managed hydrology would permit a more precise calculation of this difference, but a coarse estimate of terrestrial water storage change attributable to imported water and groundwater irrigation can be made using reported data and simplifying assumptions. From Table SJR-19 in Volume 2 of the 2013 California Water Plan Update (California Department of Water Resources,

2014) we can estimate a net export of 3.125 cm of surface water for the San Joaquin Basin for water year 2009. Assuming the ratio of consumptive use (taken to be equivalent here to ET) to applied water is 0.66 (from values in Table SJR-19), and that 7.34 cm of groundwater is extracted and applied within the basin in water year 2009, an additional 4.88 cm of terrestrial storage loss is attributable to groundwater irrigation. Taken together, these losses equal approximately 8 cm. If these losses

manifest as an absolute increase in TWS amplitude over water year 2009, which likely overstates its impact, adjusting the SJBM TWS accordingly would result in a signal amplitude only slightly larger than that given by the GRACE error bounds. Further work to constrain the impact of irrigation and water management activities on GRACE-derived TWS estimates is warranted, but the results shown here suggest that 1) comparisons between simulations of natural systems and modern TWS estimates may still be valid and useful and 2) for this study in particular, agreement between simulated and inferred terrestrial water storage change (from GRACE) provides evidence that the SJBM properly captures large-scale spatial and temporal storage dynamics.

## 5. Analysis and Discussion

The broad span of processes explicitly simulated over the entire San Joaquin watershed allows the synthesis of basin-scale hydrologic dynamics that arise from system properties and process interactions. Here we present two such analyses: 1) transient water budgets for the Central Valley aquifer and surface systems and 2) an assessment of temporal variation in watershed-scale stream-aquifer interactions.

### 5.1 Groundwater Budget

Lateral boundary conditions to the Central Valley portion of the SJBM are not predefined but are the consequence of transient solution of governing equations across those boundaries. Thus it is possible to evaluate a complete water budget for the Central Valley that incorporates physically-consistent external driving forces. Furthermore, simulated water budget components of interest, e.g. flow in the San Joaquin River, can be expressed as the combination of other budget components. Given the importance of groundwater and surface water fluxes in the San Joaquin Basin, we present an analysis of the Central Valley water budget in the context of both the saturated groundwater system and the surface system. Although we do not present results specifically for the unsaturated zone (dynamically simulated in the model as unsaturated cells below the land surface and above the water table), the surface and saturated groundwater systems are inherently tied to conditions there and thus reflect the implicit effect of the unsaturated zone.

We calculate fluxes across lateral and vertical boundaries of the Central Valley portion of the SJBM using simulated pressure and saturation. With a calculated storage change, the subsurface budget can be quantified. We limit the water budget calculations here to the bottom model layer as it encompasses our conceptualized aquifer. The aggregation of hydrostratigraphic properties and the numerical grid over this large subsurface interval means that the simulated results do not capture detailed variation in fluxes and gradients that arise from a more refined representation of heterogeneity. For this study, however, we are interested in the bulk fluxes into and out of the roughly 20,000 km$^2$ Central Valley area and consider the simplified representation of the aquifer sufficient to represent the system dynamics at that scale. In the following discussion, volumetric fluxes are presented as a depth using the area of the Central Valley within the domain (21,791 km$^2$) for the dimension conversion.

The saturated lateral flux from the Sierra Nevada and Coastal Ranges mountain blocks into the Central Valley was calculated using a head gradient consistent with the terrain-following grid formulation (Maxwell, 2013). Flux through vertically adjacent saturated cells was also computed using the Darcy equation and simulated head values. The difference between these net saturated fluxes (in minus out) and the change in storage in this model layer between time steps yields a residual

that we interpret as the net total vertical and lateral flux through unsaturated cells. Computing this water balance for each hourly time-step, aggregating to daily totals, and converting to an area-averaged depth yields the time series shown in Fig. 7. We present the water budget for the simulated Central Valley aquifer as decomposed into recharging and discharging fluxes, with a residual change in storage. It is important to note that all flow rates (m d$^{-1}$) shown in Fig. 7 are the net aggregate over a large area that may contain a mix of negative (discharging from the aquifer) and positive (recharging the aquifer) local

fluxes. The positive portion of the plot in Fig. 7 represents recharging fluxes, i.e. water that flows into the Central Valley aquifer from above or at the lateral boundary with the Coastal or Sierra mountain blocks. We interpret three recharge mechanisms from the positive portion of the graph: 1) background vertical diffuse recharge, 2) focused event vertical recharge, and 3) lateral mountain block recharge, with a total of 3.3 cm of recharge over the year. For reference, Faunt (2009) (citing Williamson et al., 1989) reports an average recharge rate of 4.67 cm for predevelopment conditions for the

entire Central Valley (including the Sacramento Basin) while Brush et al. (2013) report a simulated recharge rate of 8.7 cm for the San Joaquin portion of the Central Valley during the 1922-1929 period, although significant irrigation occurred during this time. In the plot in Fig. 7, the red filled area comprises recharge types 1 and 2 while the green area represents the third type. The background diffuse recharge is represented by the relatively flat portion of the red filled area and can be defined as the recharge flux that exists after the immediate effects of event recharge (type 2) have passed. In other words,

this recharge is the result of water that escapes below the 2-m root zone through slow drainage of surface waters or soil moisture. This recharge mechanism is identifiable as the primary vertical flux into the aquifer during the dry late summer and early fall in the Central Valley (August-November) and accounts for 1.83 cm of the total 3.3 cm of recharge.  The relative constancy of this flux suggests that it may depend on some average landscape property (e.g., soil moisture, vertical gradients, or hydraulic properties), however this remains a hypothesis for future study. The second, event-driven recharge

mechanism is apparent as the peaks and subsequent recession above the background recharge rate (type 1) in the red filled area in the plot. These peaks correspond to local precipitation and mountain runoff events that increase surface moisture and concentrate surface flows in the San Joaquin River and tributaries on the Valley floor. This increase in surface moisture increases the relative permeability of the unsaturated zone and facilitates downward movement of subsequent infiltration pulses. Runoff pulses from the mountains and local sources increase the pressure head in channels, further promoting rapid

downward flux of water. The succession of recharging pulses is followed by a more gradual recession to the background recharge rate as the saturated vertical profile in the root zone drains (and/or is depleted by ET). The area of the curve above the background recharge rate (0.005 cm/day) yields an annual recharge flux of 0.70 cm (21% of the total 3.3 cm recharge), over the Central Valley area, for this focused recharge mechanism.

An assumption common to previous numerical and conceptual models of the Central Valley aquifer system has been that the portion of recharging fluxes arising from diffuse mountain block sources are negligible in comparison to more focused mountain front (i.e. loss of flow to the subsurface where streams exit mountains onto coarse valley and fan deposits) and diffuse precipitation recharge sources (Bolger et al., 2011; Faunt, 2009; Williamson et al., 1989). Previous models follow

this assumption using a no-flow external boundary condition at the edge of the Central Valley. Notably, recent implementations of the C2VSIM model relax this assumption and include defined subsurface fluxes at the valley boundary to incorporate inflows deriving from small area watersheds in, primarily, the Sierra Nevada foothills (Brush et al., 2013). We have relaxed this assumption fully: the valley-mountain block boundary is simulated the same as every other internal cell-to-cell interface in the model. Given this, SJBM results provide a means to test the assumption that mountain block flux is a

negligible component of recharge to the Central Valley groundwater system.

The green area in Fig. 7 shows that lateral mountain block flux contributes a sizable portion of total recharge to the Central Valley aquifer in the simulation. The flux is essentially time invariant, with only a modest decline during January-March that we attribute to a reduction in gradient accompanying peak vertical recharge. The lateral recharge shown is the combination of Sierra Nevada and Coastal range subsurface flux, although the flux is predominantly derived from the Sierra side of the

Valley. Over the simulated year, the lateral flux contributes 0.77 cm (23%) of the total 3.3 cm recharge to the aquifer system. The mountain block recharge simulated in the SJBM depends on hydraulic properties of the mountain block-valley interface. Estimates of hydraulic conductivity of the mountain block (ranging from granites to continental and marine sedimentary rocks) are scarce in general, and effective regional values specific to this system are even rarer. To assess the sensitivity of lateral recharge to this uncertainty, we repeated the SJBM simulation with hydraulic conductivity for the dominant granitic

portion of the Sierra mountain block ($K_{MB}$) varied over 4 orders of magnitude from 4.2 x $10^{-6}$ m/hr to 0.042 m/hr, covering the range of variability documented for similar mountain block systems (Welch and Allen, 2014). All other simulation inputs (i.e. atmospheric forcings) are identical across the sensitivity analysis. A three-year iterative series of yearly simulations are run with each mountain block hydraulic conductivity value to allow the system to re-equilibrate to the changed subsurface property; recharge values are analyzed from the third simulated year.

The mountain block recharge flux resulting from these simulations is summarized in Table 1. Note that the recharge fraction of precipitation is taken as the total Central Valley aquifer recharge while the watershed precipitation includes the entire drainage area in the Sierra Nevada and Coast Range. The fraction of total Central Valley aquifer recharge coming from mountain block sources varies directly with the hydraulic conductivity value—with the mountain block recharge fraction ranging from 7.7% at $K_{MB}=4.2 \times 10^{-6}$ m/h to 53% at $K_{MB}=4.2 \times 10^{-2}$ m/h. However, the total amount of recharge from lateral

and vertical sources varies little for all but the largest $K_{MB}$ values, indicating that decreasing the $K_{MB}$ value partitions more water towards mountain runoff that ultimately infiltrates as mountain front or Central Valley stream loss recharge. Increasing the $K_{MB}$ value above ~$10^{-2}$ m/h however seems to substantially shift the behavior of the system: here mountain block recharge becomes the dominant recharge mechanism and total recharge amounts increase substantially. While not impossible, the high mountain block recharge fraction and attendant reduction in mountain runoff seem beyond a reasonable

conceptualization of the Central Valley and comparable systems (Manning and Solomon, 2005). Therefore we consider the range of $K_{MB}$ values from 4.2 x10$^{-6}$ to 4.2 x 10$^{-3}$ to define the envelope of variability in mountain block recharge fraction, which for this system, is 7.7% to 23%.

The balancing outgoing flux in the aquifer water budget (Fig. 7) is a net upward saturated vertical flux. Again, this is a net value representing a mix of incoming and outgoing fluxes, located in wetland and stream channel locations across the valley floor. The blue area shown can be seen as an approximation of potential groundwater contributions to stream flow. The actual stream gain is something less than this as this upward flux is partitioned into soil moisture changes (storage) and ET, as well as stream flow. Over the cumulative annual time span, this discharging flux equals a depth-normalized 3.7 cm. The 0.4 cm difference between recharge and discharge manifests as a cumulative loss of storage, indicating a small aquifer-wide drop in water table. This is the result of WY2009 being slightly drier than average and the corresponding tendency for such conditions to produce a net annual discharge (rather than recharge) for semi-arid basins like the San Joaquin.

A comparison of recharging and discharging fluxes exposes a notable feature of system function. For the portion of the water year transitioning from dry to wet, i.e. October through March, increases in recharge are accompanied by decreases in discharge. Observing at the scale of days to weeks, the recharge event peaks (red) in this time period are preceded by a depression and subsequent rebound of discharges (blue). The general relationship extends from March through mid-May, but over this period the peak recharge pulses recede and the discharge flux rises, responding to previous months' recharge. Finally, in the dry summer months (June-September), both discharge and recharge recede in concert toward their baseline levels. We attribute this overall phenomenon to the temporary reduction or reversal of vertical stream-aquifer gradients during runoff events and the subsequent increase in gradients as the surface dries and recharge redistributes. This asynchronous behavior plays a role in watershed-scale groundwater-surface water interaction evolution that will be discussed in a later section.

**5.2 Land Surface Budget**

The groundwater budget shows, in general terms, the connections between the groundwater and land surface systems. The variety of processes active at the surface – interception, transpiration, evaporation, overland flow, and more extensively occurring variably saturated flow –means the water partitioning process in that domain is more complex. In order to couch this complexity in terms of a more relevant and concrete result, we present a transient decomposition of the water budget at the land surface in terms of the simulated Central Valley surface outflow hydrograph. For the SJBM, this hydrograph is composed of flow from the Kings River and the San Joaquin River, although the flow from the latter is the predominant component.

The shaded portions of the plots in Fig. 8 and Fig. 9 show a transient incremental and cumulative simulated surface water budget, aggregated to 3-day increments for clarity. As before, water budget fluxes are shown as a depth using the area of the simulated Central Valley as the normalizing factor. The black line in these plots represents the Central Valley outflow and is the residual of the positive (above axis) and negative (below axis) shaded water budget components. The budget shown is

calculated for the first model layer and the corresponding overland flow and surface storage. Inflows to this domain of interest are fluxes that can be considered to add to stream flow and include: upward saturated vertical flux into stream channels and wetlands (referred to as 'stream gain'), precipitation (more precisely canopy throughfall over vegetated cells), inflow from streams at the valley edge, vertical unsaturated flow, and release of stored soil water in the top model layer.

Likewise, the outflow components are fluxes that reduce streamflow, and include downward vertical saturated flux through stream channels ('stream loss'), unsaturated downward flux past the top model layer, ET, and an increase in stored soil water. While not strictly a flux, transient soil moisture storage changes, separated here into gains and releases, is critical to closure of the water budget. One can think of a storage gain as water that is temporarily stored in the surface soil but is otherwise immobile – thus not contributing to streamflow (or other outgoing flux) –*at that time step*. Conversely, a storage

release is a reduction of soil moisture that can contribute to any coincident outgoing flux.

The surface budget plots highlight several aspects of the predevelopment Central Valley hydrologic system. First, the cumulative plot shows the valley inflows dominate the surface water budget, effectively equaling the resulting valley outflow. Yet, the presence of the other budget components demonstrates the outflow hydrograph is the product of more interaction than a simple downstream translation of mountain streamflow. Valley floor stream losses and gains are

approximately equivalent (3.47 cm and 3.13 cm, respectively) in magnitude over the year simulation and, singly, represent more than 10% of outgoing streamflow, suggestive of an active exchange of surface water and near-surface saturated groundwater in predevelopment riparian zones. As described earlier, the vertical fluxes from the bottom model layer (the approximated aquifer system) are related but not wholly attributable to the surface stream-aquifer exchange fluxes. The net upward flux out of the groundwater contributes variably to ET and soil moisture changes as well as streamflow. While at

times dwarfed by precipitation and mountain inflow fluxes, this groundwater flux is a significant component of the surface budget during the dry summer and fall months in the valley.

The incremental time-series plot shows that the Valley surface outflow is a result of competing additive and subtractive processes. This time series can be broadly divided into two seasons: a first, local precipitation-dominated regime spanning October-March, and a second, mountain stream inflow dominated regime, spanning March-September. In the first regime,

positive fluxes are dominated by local precipitation events but with little relative increase in streamflow. The surface budget components show that this water, and sometimes an accompanying pulse of mountain streamflow, is partitioned first into storage and root zone infiltration, then subsequently partitioned toward storage releases, ET, and streamflow. In the second regime, little precipitation falls on the Central Valley relative to snowmelt-driven mountain inflows. These inflows encounter a Valley floor already wetted by the preceding winter rains and thus contribute less to maintaining soil moisture or diffuse

recharge but rather constitute the bulk of the ET flux, stream losses, and valley stream outflows.

To better illustrate how the surface budget components relate to the simulated stream outflow, we distribute the net positive fluxes across the outflow hydrograph, yielding a water budget-based hydrograph separation. This separation is only one possible interpretation possible from the simulated water budget values and does not take into account process pathways that may affect how relative proportions of different sources and fluxes contribute to the downstream hydrograph. Rather, this

approach assumes stream gains always contribute to downstream flow and that precipitation, soil storage release, and valley inflows contribute to the stream outflow in direct proportion to their fraction of total positive fluxes (excluding stream gains). A more detailed accounting would require simulation and processing capabilities beyond the scope of this study but remain an important topic for future work. The separated daily hydrograph in Fig. 10 (left) reveals the dynamic composition

of streamflow through precipitation events and seasonal snowmelt. First, the saturated groundwater contribution (orange), or, more simply, baseflow, is relatively invariant but comprises a large portion of summer and fall streamflow leaving the simulated Valley. Mountain stream inflows (red) and local precipitation (blue), in contrast, are more variable and tend to constitute the rising limbs of each event hydrograph spike. In this case, the precipitation component in the hydrograph can be interpreted as local runoff generation that is routed to a Central Valley outlet. After precipitation ends, the variable

components of streamflow switch to storage release (green) and mountain stream inflows, with an increasing portion of the recession flow dominated by release of water from storage in the surface system as saturated soils or ponded water drains. To the extent that some portion of this stored water remains in place (i.e. is not released) between events, the storage releases (green portion of the hydrograph) represent an estimate of the surface portion of so-called "old water" (Kirchner, 2003; McDonnell, 2003). As we consider only the surface layer in the model, contributions to streamflow from lower layers are

necessarily incorporated in the baseflow (orange) component. The assumption that each flux type contributes to streamflow proportionally to its part in the total positive flux is difficult to assess against observation. However, given that the outflow hydrograph cannot be fully constructed from a uniform allocation of certain fluxes across the simulation period, the proportional separation shown here seems a reasonable first approximation. Validation of this approach again points to an important topic for future consideration.

Much like in the deeper simulated aquifer system, hydrologic processes at the land surface act to transform the inflow time series to an outflow hydrograph, but the nature of this transformation varies through an event and across seasons and depends on the antecedent conditions of the region. Viewed another way, this temporally varying inflow transformation implies a changing importance or dominance of processes in time. We examine this through a spectral analysis of the hourly components of the separated hydrograph, shown as a periodogram in Fig. 10 (right). Filtered versions of each component

(heavy lines) are overlaid on the raw power spectra for ease of interpretation. The plot demonstrates a range of temporal scaling behaviors for the hydrograph components: precipitation exhibits the expected approximate white noise (no temporal correlation), streamflow hydrographs into and out of the valley demonstrate a complex scaling structure with a transition between two fractal dimensions at a period of 12-24 hours, and the storage and stream gain components scale more or less uniformly with period, with notable local peaks at the 12 and 24 hour periods. The general shape and relative arrangement of

these component spectra is qualitatively consistent with previous studies (Zhang and Schilling, 2004), but the high (hourly) resolution and limited one year time series prevent a more in depth analysis of potential changes in scaling behavior from the monthly to multi-annual time scales that is more commonly examined. The simulated time series does allow, however, some preliminary inferences about hydrograph contributions at the diurnal to monthly scale. The similar forms of the Valley inflow and outflow hydrograph spectra suggest multiple scales of temporal correlation in streamflows, a strong persistence

or correlation over periods of hours to days (indicative of storm events and dominant diel fluxes like ET), and a slightly diminished correlation over the scale of weeks to months (representing the strong seasonal precipitation patterns in the Mediterranean climate of the San Joaquin Basin). Furthermore, the similarity of the outflow and inflow hydrograph spectra suggest that much of the temporal scaling in the Valley outflow is largely established by runoff and routing processes in the

bounding mountain systems, predominantly the Sierras. The effect of the Central Valley hydrologic processes on the streamflow manifests as a subtle straightening of the slope of the outflow spectrum over the 12 h to 5 d range, perhaps indicating a stronger connection between surface water and groundwater that makes smoother the transition between, for example, vadose zone dominant and shallow groundwater dominant scaling regimes (Thompson and Katul, 2012). Notably, all spectra except that of the stream gain component converge toward similar (essentially uncorrelated) power values at

periods greater than ~12 days. The continued scaling of the stream gain component over this range suggests an important long-term control over streamflow variability for this system.

### 5.3 Watershed groundwater-stream interactions

The groundwater and land surface water budgets reveal that the processes that produce and modify streamflow across the Central Valley vary in their frequency and intensity over the simulated water year. In particular, these analyses suggest a

balance of fast and slow response mechanisms at the surface and in the groundwater, with the unsaturated connection between the two serving to introduce a time lag and filter high frequencies while the direct saturated connection (often in stream channels and wetlands) allows fast response through pressure changes. Given the variability that results from these heterogeneous streamflow generation and modification mechanisms, one would expect a nonlinear relationship between connected groundwater and surface water systems. We examine one aspect of this nonlinearity: the hysteretic relationship

between groundwater contributions to streamflow and total stream discharge.

Figure 11 shows the path marked by stream gain and streamflow volumes for every hour of the one-year simulation of the San Joaquin River (including the mountain tributaries). The color scale denotes the temporal component, with blue and red colors indicating early and late simulation time, respectively. Although somewhat obscured by the high frequency variations in stream gain and watershed runoff, the points cluster around a path that traces a clockwise hysteretic loop that begins

(October 1) in a regime of low streamflow and low stream gain, increases toward a regime of high streamflow and high stream gain as the watershed wets under winter rainfall and late spring snowmelt following a semi-log concave-down curvilinear path (through ~May), and then returns to the original state following a concave up, semi-log curvilinear path.

This path is, in general, consistent with similar patterns relating runoff to storage at large regional (Riegger and Tourian, 2014; Sproles et al., 2015) and field and site scales (McGlynn et al., 2004; Spence et al., 2010). We extend from these

previous studies to examine the groundwater-streamflow relationship more specifically; the schematic at right in Fig. 11 presents an interpretation of the processes that contribute to this annual hysteretic loop. Evapotranspiration reduces or reverses gradients that drive water into streams and reduces stream gains, a flux that drives the annual curve in a negative x-direction. The SJBM simulation shows the effect of ET to be generally large, but over short (diurnal) duration. Interestingly,

the stream gain reducing impact of ET seems to be bounded by a roughly log-linear slope as a function of stream gain. We interpret this to be a result of the correlation between seasonal reduction in potential ET in winter and spring and the high stream gain/high stream flow regime that would naturally accompany it for this system. Basin-scale recharge to groundwater and lateral converging flow counteracts the force of ET, supporting discharging gradients of groundwater to streams and

forcing the annual curve in positive x-direction. The shape of the rising limb of the annual loop is then partially a result of the changing rate at which recharge and lateral flow overcome the effects of ET. Changes in the y-direction (basin streamflow) are driven over short time scales by precipitation events that produce pulses of overland flow. In cases where the precipitation event facilitates recharge to or propagation of a pressure increase through the saturated subsurface, the recession of the runoff pulse deflects in the positive x-direction, a feature apparent in runoff events in the January-March

range in Fig. 11 (left). Thus, the contribution of groundwater to total streamflow exerts a longer-term control on the curve in the vertical direction. This extends to the seasonal recession of flows that occurs during the seasonally dry June-October time period: without precipitation or snowmelt inputs, streamflow is maintained by an increasing fraction of groundwater contributions, which in turn control the decrease in dry season streamflow.

**6 Summary and conclusions**

We present an integrated hydrologic model of the San Joaquin River basin that simulates the full terrestrial water budget for an approximation of natural, predevelopment hydrologic conditions. Conceptualized such that differences in mountain and Central Valley hydrology arise as the result of differences in hydraulic and land surface properties rather than a modification of the underlying governing equations, the SJBM accurately captures important hydrologic phenomena such as groundwater gradients in the Central Valley, partitioning of precipitation into runoff and ET, and seasonal variations in terrestrial water

storage. Mismatches between simulated and observed water budget components, notably in absolute streamflow volume and snow water equivalent, can be attributed to precipitation underprediction over the complex Sierra Nevada terrain in the reconstructed meteorology product, highlighting the need for improved forcing products to support high-resolution hydrologic modeling for similar systems.

Analysis of a one-year SJBM simulation reveals the transient interactions among and within key portions of the San Joaquin

hydrologic system. A decomposition of fluxes to and from the Central Valley aquifer system shows a lag between peak recharge and discharge within the annual cycle. Furthermore, the representation of the Valley-mountain block interface as a regional continuum shows that a small but temporally constant portion of the recharge to the Valley aquifer comes as lateral flow from the mountain block. Considering the uncertainty in mountain block hydraulic conductivity, mountain block recharge as simulated in the model varies between 7% and 23% of total Central Valley aquifer recharge. These values

represent estimates of system characteristics relevant to questions of change from predevelopment to modern conditions.

The multiple components and high temporal variability in the surface system can easily obscure the connections between groundwater, surface hydrologic components, and resulting streamflow. We provide a simulated water budget-derived

hydrograph separation for Central Valley river outflow that reveals the variably dominant sources of water that produce and maintain surface flow across the Valley; local precipitation is important for rising hydrograph limbs but much of the streamflow volume is sustained by subsequent Valley inflows and release of stored soil and surface water while direct groundwater contributions are comparatively steady throughout the year. Power spectra of the hydrograph components show

that, for the natural system, temporal patterns in streamflow are largely set by inflows from mountain runoff while scaling of baseflow contributions suggest a possible longer-term effect of the Valley.

Finally, the SJBM reveals an annual hysteresis inherent in the groundwater connection to streamflow across the San Joaquin River basin. The hourly simulation results permit a detailed resolution of the path traced through the dry-wet cycle of the basin and forms the basis of a conceptual interpretation of the process contributions to this phenomenon.

This study demonstrates the utility of an integrated hydrologic modeling approach to reveal interactions among terrestrial hydrologic components and, more importantly, the role of these interactions in affecting the observable hydrology of regional systems. As an approximation of natural or predevelopment conditions, the simulated results provide a reference state relevant to questions of human impacts on this crucial hydrologic system. In addition, this study provides insight into the mechanistic underpinnings of groundwater-surface water connections at a regional scale that are potentially applicable to

basins beyond the San Joaquin: properties that control the contributions to the streamflow hydrograph and the nature of basin-scale streamflow-groundwater hysteresis may allow inference about similar behaviors in a more general setting. However, future observational and simulation research is needed to address important aspects not sufficiently resolved here such as: characterization of constant and transient factors affecting groundwater-surface water hysteresis, validation of riparian vertical gradient variation in time and space for the San Joaquin and similar systems to refine modeling

representation of such processes, and investigation of how the modern, impacted systems change the "natural" system results presented.

**Code and Data Availability**

Simulation inputs, models, and data are archived and are available from the authors upon request. Specific versions of the ParFlow model source code are archived with documentation and may be downloaded at

(http://inside.mines.edu/~rmaxwell/maxwell_software.shtml) or checked out from a commercially hosted, free SVN repository and GitHub; version r895 was used in this study.

**Conflicts of Interest**

The authors express no conflicts of interest.

**Acknowledgements**

We would like to acknowledge high-performance computing support from Yellowstone (ark:/85065/d7wd3xhc) provided by NCAR's Computational and Information Systems Laboratory, sponsored by the National Science Foundation. Additional computational resources were provided by the Golden Energy Computing Organization at the Colorado School of Mines using resources acquired with financial assistance from the National Science Foundation and the National Renewable Energy Laboratory. This research was supported by funding from the National Science Foundation through its Climate Change, Water, and Society (CCWAS) Integrative Graduate Education and Research Traineeship (IGERT) program (http://ccwas.ucdavis.edu/; DGE-1069333).

# Appendix

## Appendix A. Model Grid

The model grid (220 rows x 270 columns) for this study utilizes a terrain-following grid transform (Maxwell, 2013) that allows a non-uniform (but constant for a given layer) vertical dimension but requires a regular horizontal (x and y dimension) discretization, set at 1000 m. The 500-meter thickness of the simulated subsurface is discretized into five layers. The top four layers have thicknesses of 0.1 m, 0.3 m, 0.6 m, and 1.0 m, respectively, and together constitute a 2.0-meter thick "root zone" intended to capture near-surface biophysical and hydrologic dynamics. The bottom layer thickness is 498.0 m and is intended to capture, as a bulk approximation, unsaturated and saturated flow below the root zone. Defining the vertical discretization in this way reduces the computational demand of the simulation but also reduces the degree to which the effects of aquifer heterogeneity, especially in the vertical direction, can be resolved. For example, the semi-confining Corcoran clay (Faunt, 2009; Williamson et al., 1989) is not explicitly represented by distinct hydraulic properties in the SJBM. This formulation precludes use of this particular model for simulation of vertical pressure gradients, especially those that are associated with transient multi-level groundwater pumping and recovery. The generalized discretization is justified in this application, though, as it captures the regional-scale horizontal gradients that drive the bulk of the saturated subsurface flow and allows continuous and integrated simulation of mountain block and Central Valley subsurface flow.

## Appendix B. Subsurface Properties

Subsurface property inputs to ParFlow include porosity, saturated hydraulic conductivity, specific storage, and soil water characteristic and relative permeability curve parameters. Input values for the SJBM come from a combination of previous modeling, data collection efforts in the basin, and reference values. Details regarding data sources and modifications of values for implementation in the SJBM are described in the following paragraphs.

The hydrostratigraphic conceptualization is mapped to model input via a three-dimensional array of integers corresponding to the grid dimensions, with each integer indicating a hydrostratigraphic category. Categories were assigned hierarchically with the domain being first divided into general geological categories (mountain block, deep valley sediments, Corcoran clay, shallow valley sediments, coastal range) and then into textural subcategories (sand, silt, clay, etc) (Mansoor, 2009). Indicators for the Central Valley portion of the domain were derived from the borehole datasets compiled by Faunt (2009) and mapped to the model grid using the most frequent texture category over a given interval. Subsurface indicators in the portion of the coastal range and Sierra Nevada mountains in the domain were considerably simpler, with textural subcategories only assigned in the near surface where SSURGO maps provided guidance

(Soil Survey Staff, Natural Resources Conservation Service, 2009). Figure B1 (a) provides an illustration of the spatial variability of the subsurface indicator categories. Note that for clarity the layers are shown with uniform thickness and with a vertical exaggeration factor of 100.

An additional hydrostratigraphic indicator category was added to allow heterogeneity in hydraulic properties within mountain valleys due to past glacial and fluvial activity. This indicator category was assigned to perennial stream channels in the bottom (498-m thick) model layer. Although this configuration overestimates montane alluvial valley sediment thickness over the portion of the Sierras in this model domain, it was considered a reasonable first approximation to capture the effects of alluvial and glacial fill known to exist as thicknesses of several hundred meters in domain river valleys like that of the Merced (Gutenberg et al., 1956).

Saturated hydraulic conductivity values were mapped to indicator categories from the values given previous studies for the Central Valley subsurface (Faunt, 2009; Mansoor, 2009). Mountain block hydraulic conductivity values were assigned based on the range given in published observational and modeling studies, like those summarized by (Welch and Allen, 2014), for example. We assigned a vertical anisotropy factor (Kz/Kx,y) for Central Valley hydraulic conductivity according to the indicator categories in Mansoor (2009). We assigned mountain block hydraulic conductivity to be isotropic. The distribution of saturated hydraulic conductivity values (m h$^{-1}$) is shown, by layer, in Figure B1 in panels (b), (c), and (d). Layers 1-3 are shown together because hydraulic parameters are assigned uniformly over the top one meter of the domain.

The van Genuchten equations define the saturation-pressure and saturation-relative permeability relationships necessary for variably-saturated subsurface flow simulation (van Genuchten, 1980). The van Genuchten parameters for residual saturation, α, and n were assigned in the model based on the textural class of each indicator category using values reported in (Schaap and Leij, 2000), with n values constrained to a minimum of two to ensure sufficient smoothness in the constitutive relations for robust solution of Richard's equation (Miller et al., 1998). The maximum saturation for the van Genuchten parameterization was assumed to be equal to one.

Subsurface storage properties are assigned through porosity and specific storage parameter values. Porosity is spatially variable according to the hydrostratigraphic indicators following values in (Faunt, 2009; Mansoor, 2009). Specific storage values are assigned a uniform value of $1 \times 10^{-5}$ m$^{-1}$ across the model domain.

**Appendix C. Land Surface Properties**

Land surface property inputs to ParFlow and Parflow.CLM control the simulated behavior of overland flow and land-atmosphere water and energy fluxes (Kollet and Maxwell, 2006, 2008). A surface slope and Manning's roughness value are required for overland flow routing at each cell. Land cover types and associated biophysical parameters are required for simulation of water and energy fluxes at the land surface via CLM. The details of these simulation inputs are provided in the following paragraphs.

The ParFlow overland flow boundary condition requires a non-zero slope at each cell to route water via the kinematic wave approximation (Kollet and Maxwell, 2006). We derived slopes from a 1/3 arc-second (~10 meters) DEM aggregated to the one kilometer resolution grid using the watershed analysis tool in GRASS GIS (Ehlschlaeger and Metz, 2014; Gesch et al., 2002; Gesch, 2007) and following the general protocol as described in Barnes et al., (2016). These slopes were further

processed to retain only the highest slope in the x- or y-direction (leaving the other direction with a zero slope) while enforcing a slope range of 0.00001 to 0.5. Slopes in model cells designated as a channel by the watershed analysis algorithm were assigned an average sub-basin slope value to maintain a continuous downstream gradient. A visual check of slope continuity was performed by applying precipitation pulses to the ParFlow domain built with these slopes but having only overland flow active (i.e. no subsurface flow).

The CLM land surface model requires as input for each model cell the vegetative cover type and corresponding biophysical parameters or properties. The vegetation type was assigned according to the most common IGBP land cover type within each cell using the MODIS land cover product (MCD12Q1) (Oak Ridge National Laboratory Distributed Active Archive Center (ORNL DAAC), 2014). Default vegetation parameters were used as input for all vegetation types except the evergreen needleleaf forests, evergreen broadleaf forests, and woody savannahs. We reduced the maximum and minimum leaf area

index for these vegetation types to address high canopy water storage and transpiration anomalies apparent in initial simulations. The maximum (minimum) LAI values were adjusted to 5 (4), 5 (4), and 4 (1) for the evergreen needleleaf, evergreen broadleaf, and woody savannah land cover types, respectively.

Manning's roughness values were assigned to each surface cell in the model domain according to criteria based on vegetation type, physiographic region, and whether the cell was likely to be a perennially wetted channel. The connected

stream segments derived from the watershed analysis algorithm designated channel cells. Channel cells were assigned a type based on physiographic province (Fenneman and Johnson, 1946). These channel types represent the general character of stream channels that might be expected and allow a means for interpreting and assigning channel roughness values. Manning's roughness values were assigned to each land cover and channel type based on reference values (Chow, 2009).

Meteorological Forcing

We performed initial PF.CLM simulations using the NLDAS-2 atmospheric forcing product (Xia et al., 2012) because it provided the consistent regional coverage at hourly resolution for the eight variables needed by the CLM land surface model in the PF.CLM code. These simulations were characterized by high ET fluxes, low snowpack, and low streamflow. Comparisons to meteorological stations across the domain suggested the forcing temperature was biased high while the precipitation was biased low, with the biggest differences occurring where the 0.125° x 0.125° NLDAS-2 grid cells

overlapped adjacent mountain ridges and valleys. Such inconsistencies are not surprising given the challenge and documented biases associated with development of gridded meteorological products over complex and sparsely instrumented terrain (Mo et al., 2012; Pan, 2003; Xia et al., 2012, 2016).

To better represent the variability of the near-surface atmospheric conditions we instead utilized a data product that synthesizes the GOES Surface and Insolation Products (GSIP), Stage-IV precipitation product, and the NLDAS-2 forcing

into a downscaled four-kilometer resolution product (M. Pan, personal communication, June 11, 2015). The downscaling process starts with adjustment of the NLDAS-2 air temperature using a constant lapse rate of 6.5°C/km and a 4-km resolution elevation dataset. Once adjusted, the new temperature is used to calculate a corresponding relative humidity that is bilinearly interpolated to the 4-km grid and converted back to specific humidity. Downward longwave radiation is also

calculated using the adjusted air temperature and the Stefan-Boltzmann law. Incoming shortwave radiation is mapped to the 4-km grid from the GSIP surface insolation dataset with an additional solar angle adjustment. The NLDAS-2 Precipitation fields are replaced with the Stage-IV product (Climate Prediction Center and Joint Office for Science Support, 2000) where available. Finally, wind speed and atmospheric pressure are bilinearly interpolated from the NLDAS-2 product to the 4-km grid.

Simulations performed using the 4-km forcing product that included the Stage-IV precipitation over the SJBM domain showed improvement in some model responses but overall were hampered by a consistent and very low estimate of precipitation in the Sierra Nevada Mountains. This is believed to be a result of 1) poor radar performance over complex and high terrain and 2) inconsistencies in the regional algorithms that produce the Stage-IV product. Given these issues, the original NLDAS-2 precipitation field was used in combination with the humidity, pressure, temperature, radiation, and wind

variables from the improved 4-km dataset.

**Appendix D. Spin-up and Model Initialization**

The SJBM was initialized dry and equilibrated to quasi-modern hydrologic conditions through a multi-stage spin-up process. First, a mean historic potential recharge flux (precipitation minus evapotranspiration with negative values set to zero), derived from the products developed by (Maurer et al., 2002), was applied to the land surface of the dry domain. This

constant flux was applied to the land surface in a ParFlow-only simulation using increasing time-steps. Due to the depth and initial dryness of the domain, the propagation of pressures from water applied at the surface throughout the full vertical profile was relatively slow. To speed this process, additional flux was added to the bottom layer and continued until surface saturation of channel locations.

After initiation of the surface flow system and development of a reasonable water table configuration in the ParFlow-only

simulation, the system was further equilibrated with dynamic forcing using the coupled land surface capabilities in ParFlow-CLM. Simulations in this stage used hourly time steps and spatially distributed and temporally variable meteorological forcing. One-year simulation iterations were performed using the same forcing, each year starting with the ending state of the previous year's simulation. This process was repeated until the total domain storage change between the beginning and ending of a simulated year was less than 1% of the initial storage for that year, following the general spin-up guidance

outlined in previous studies (Ajami et al., 2014; Kollet and Maxwell, 2008; Rihani et al., 2010). Additional qualitative checks of snowpack, streamflow, and land surface temperatures were also performed to ensure that large year-to-year variations had been minimized at the end of the spin-up process.

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

**Table 1.** Sensitivity of mountain block recharge in the San Joaquin Basin to Sierra Nevada mountain block hydraulic conductivity

| Mountain block hydraulic conductivity | Mountain block recharge (MBR) [cm]* | Total Central Valley recharge [cm]* | MBR recharge fraction [-] | Total valley recharge as fraction of watershed precipitation [-] |
|---|---|---|---|---|
| $4.2 \times 10^{-6}$ m hr$^{-1}$ | 0.196 | 2.55 | 0.077 | 0.021 |
| $4.2 \times 10^{-4}$ m hr$^{-1}$ | 0.292 | 2.58 | 0.113 | 0.022 |
| $4.2 \times 10^{-3}$ m hr$^{-1}$ | 0.770 | 3.30 | 0.231 | 0.028 |
| $4.2 \times 10^{-2}$ m hr$^{-1}$ | 2.614 | 4.91 | 0.532 | 0.041 |

*Normalized to simulated Central Valley area

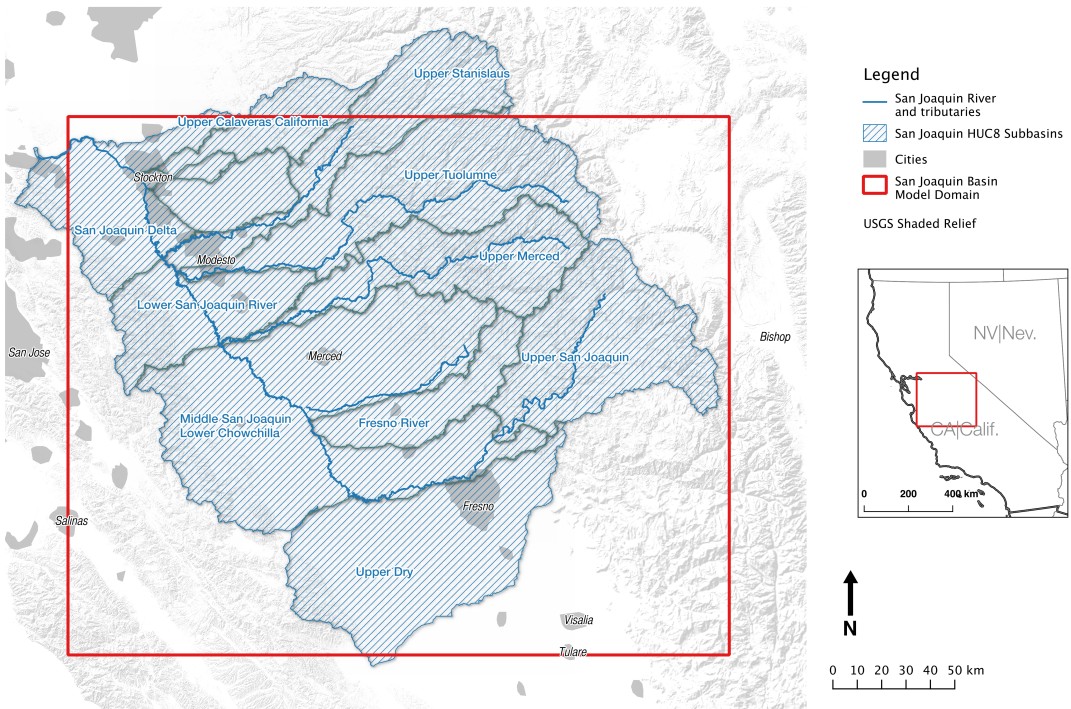

5    **Figure 1: Extent and configuration of the San Joaquin Basin Model study area. The red box defines the 59,400 km² ParFlow-CLM domain. The 1-km model grid is 270 columns by 220 rows. The model domain covers the majority of the surface drainage (hashed blue polygons) contributing to the San Joaquin River and its main tributaries (blue lines).**

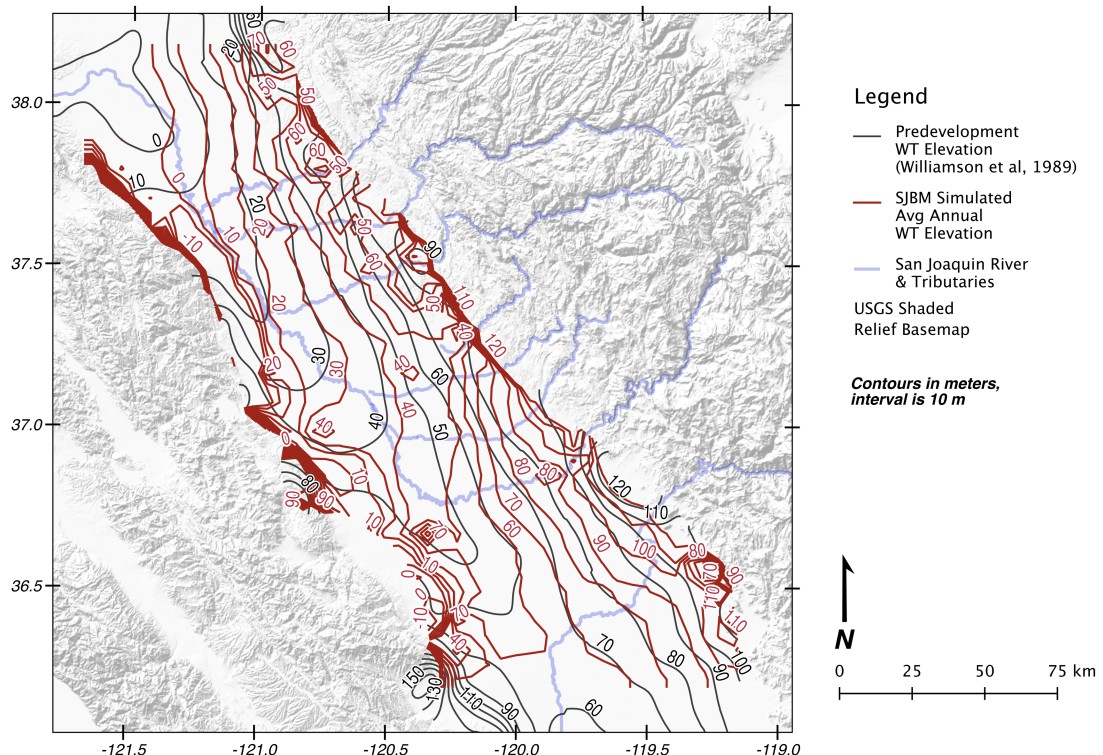

**Figure 2: Predevelopment water table comparison. Red contours are mean annual effective water table elevation (m) as simulated by the San Joaquin Basin Model. Black contours are estimated predevelopment water table elevation (m) converted from Williamson et al, 1989.**

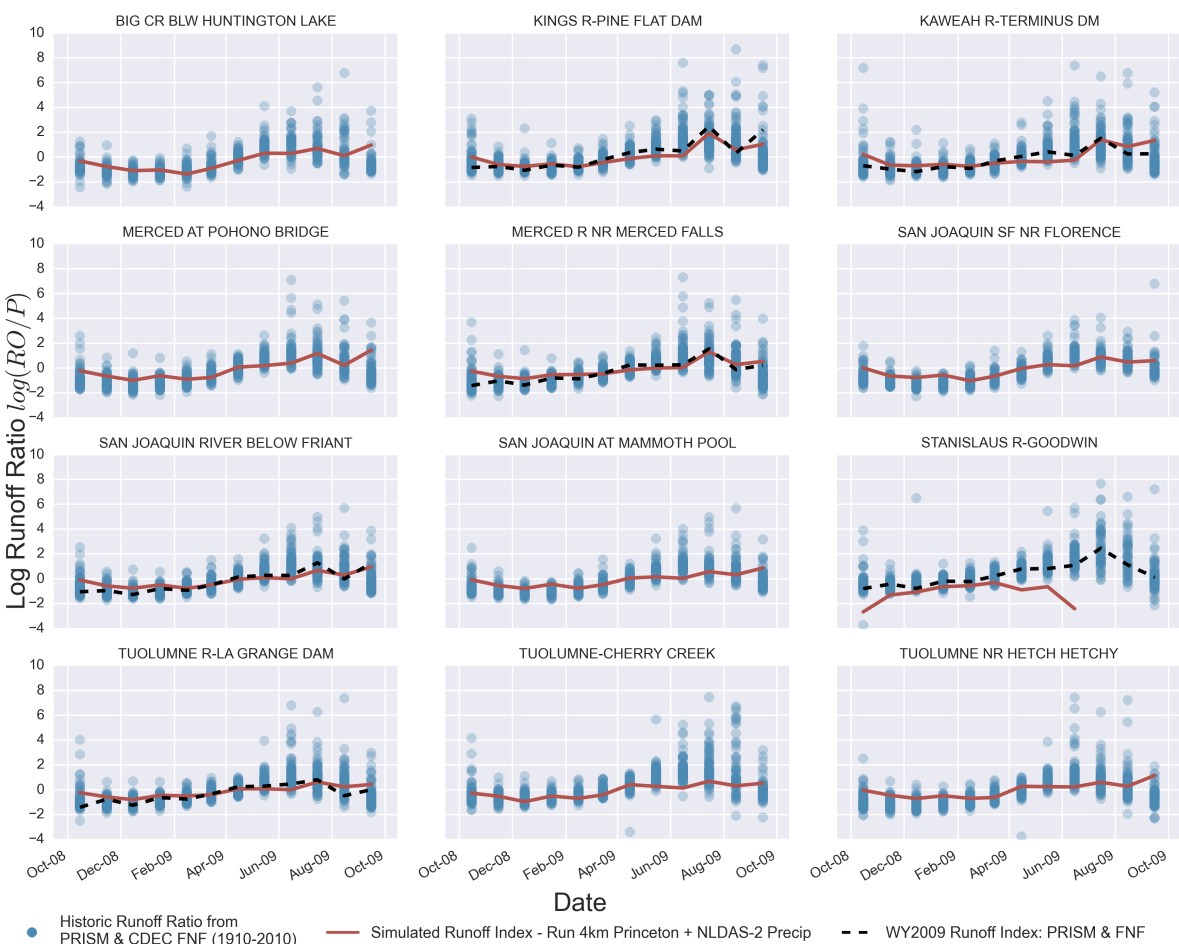

**Figure 3: Monthly runoff ratio (streamflow volume as a fraction of monthly precipitation volume. Blue dots represent available historical record from 1910-2010 using PRISM precipitation product and the California Department of Water Resources estimated full natural flow (FNF) for 12 watersheds with gages in the San Joaquin River Basin model domain. Simulated runoff ratio is shown as a red line. The dashed black line indicates the historical runoff ratio for water year 2009 where FNF station data existed.**

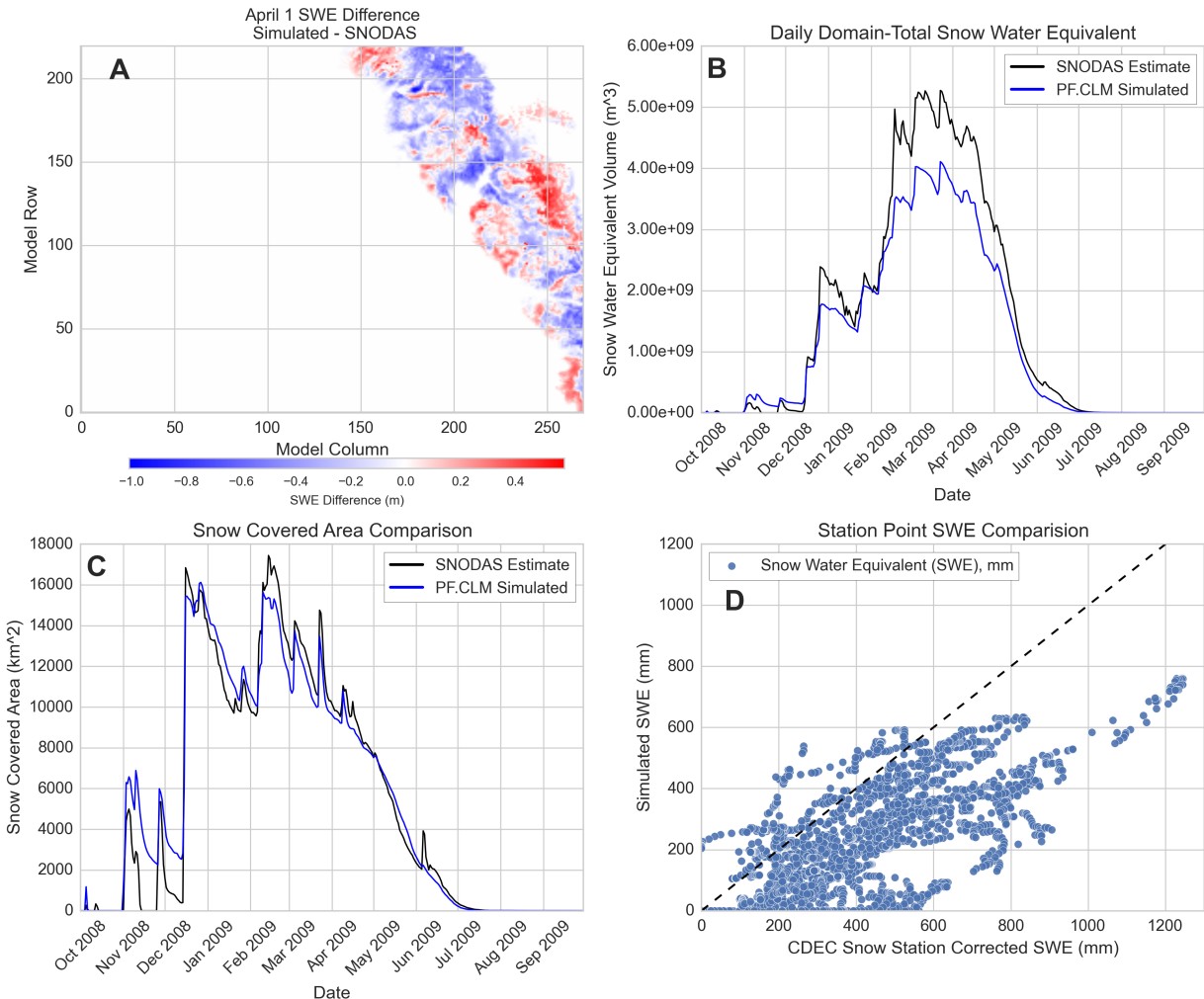

**Figure 4: Comparisons between simulated and observed snow water equivalent (SWE) snow covered area. (A-top left) SJBM simulated April 1st SWE minus SNODAS product. (B-top right) Daily domain total simulated (SJBM) and observed (SNODAS) SWE. (C-bottom left) Daily domain total simulated (SJBM) and observed (SNODAS) snow covered area. (D-bottom right) Point SWE comparisons for winter 2008-2009, simulated (SJBM) versus observed (California Department of Water Resources snow stations).**

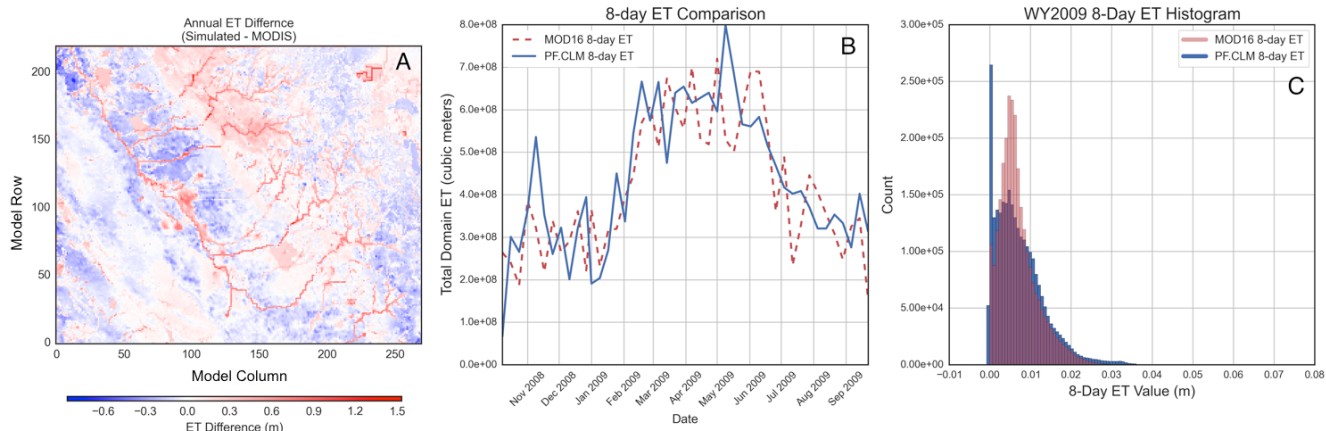

**Figure 5: Comparison of simulated evapotranspiration (ET) to MODIS estimates over the SJBM domain. (A) Annual difference in ET (m) is shown spatially. Note the high simulated ET in channels and riparian zones not captured in the MODIS product. (B) Time series of spatially-averaged 8-day ET values for SJBM and MODIS show good agreement. (C) Histogram of 8-day ET values shows higher variance in SJBM simulation.**

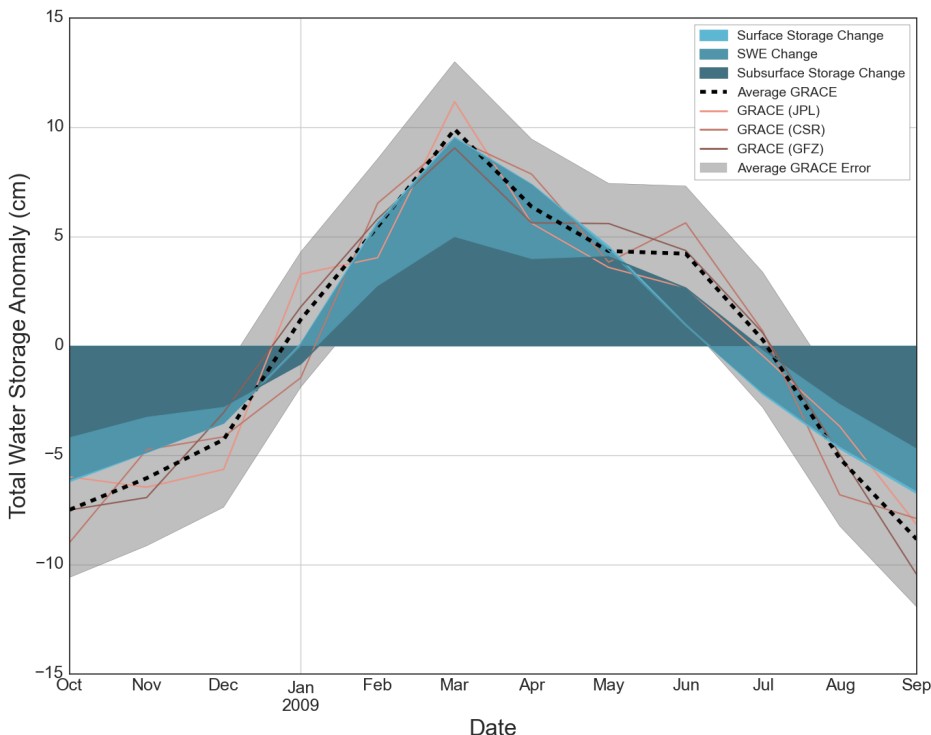

**Figure 6: Total terrestrial water storage (TWS) comparison between GRACE (dashed, solid lines, grey area) and the San Joaquin Basin Model (blue filled areas). Simulated annual amplitude in storage change is lower than that derived from the GRACE signal. This discrepancy can be attributed to precipitation shortfalls in the forcing dataset and lack of pumping, irrigation, and surface retention in the simulation.**

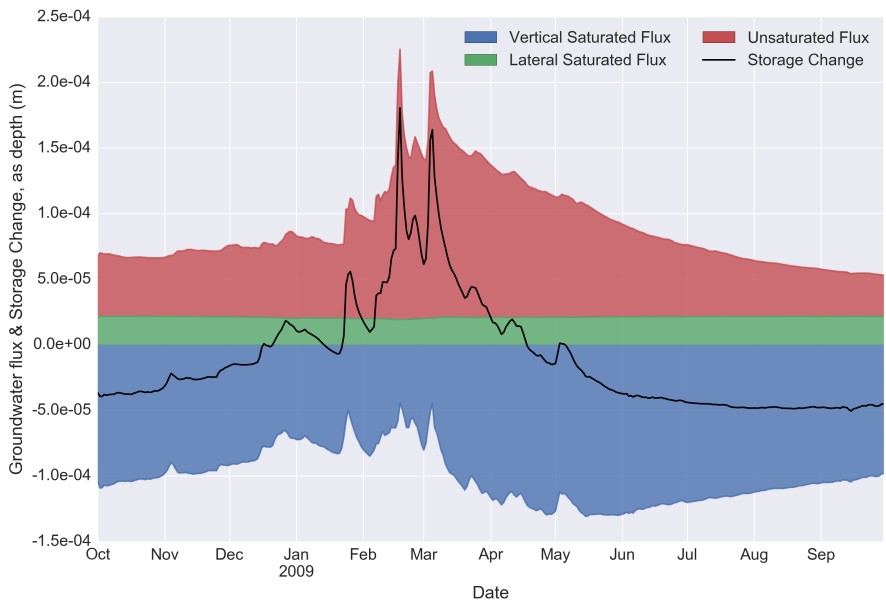

**Figure 7: Daily time series of bulk fluxes to/from the simulated Central Valley aquifer system. Positive values indicate a net flow into the aquifer. Negative values indicate net flow out of the aquifer. The balance of inflows and outflows is shown as the change in storage (black line).**

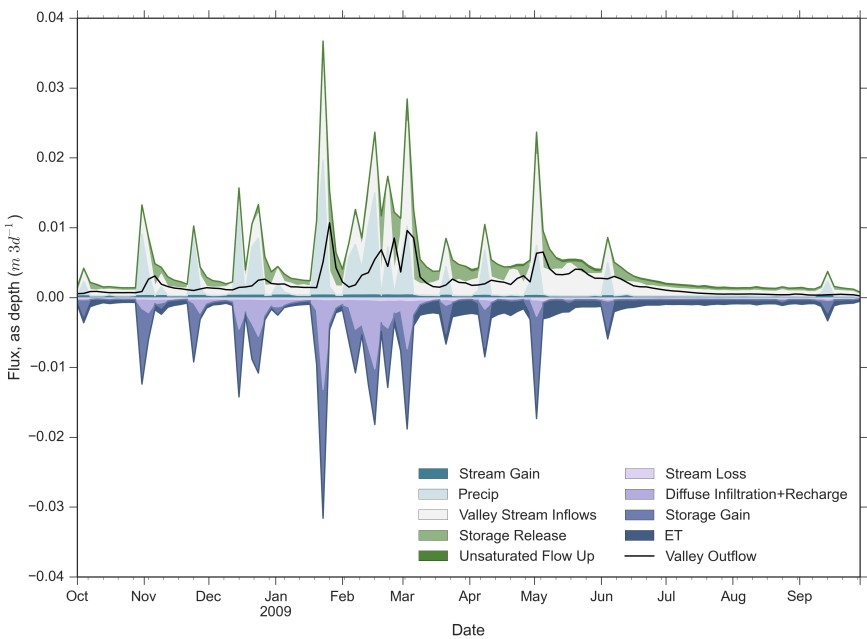

**Figure 8. Three-day simulated surface water budget time series, presented as components of the resulting outflowing stream hydrograph. Flow values shown as depth in meters per three-day period, normalized by simulated Central Valley area. Positive values represent flow that potentially contributes to streamflow. Negative values represent flow that potentially reduces streamflow.**

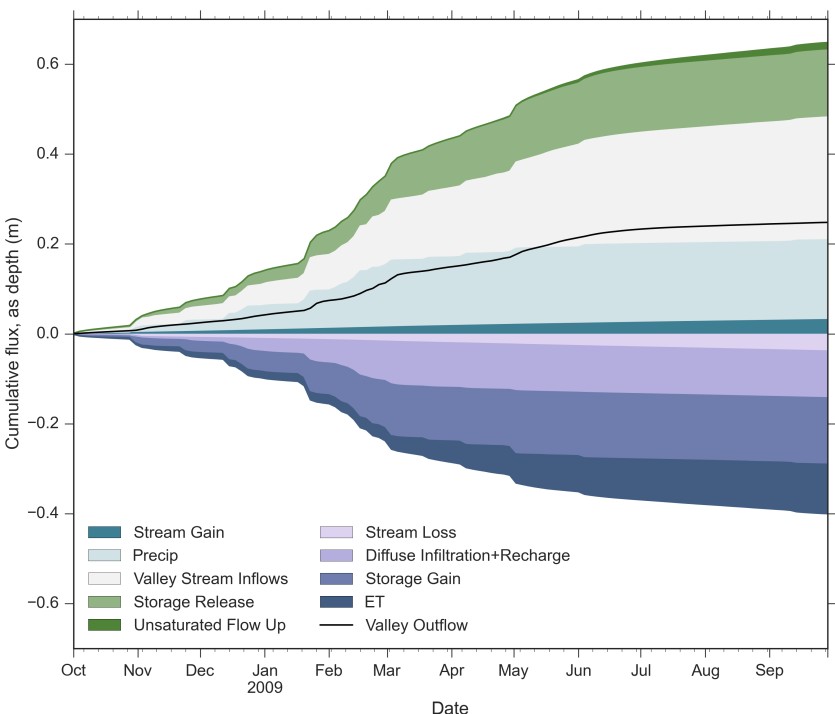

**Figure 9: Cumulative simulated surface water budget time series as components of the resulting outflowing stream hydrograph. Values shown as depth in meters, normalized by simulated Central Valley area. Positive values represent flow that potentially contributes to streamflow. Negative values represent flow that potentially reduces streamflow.**

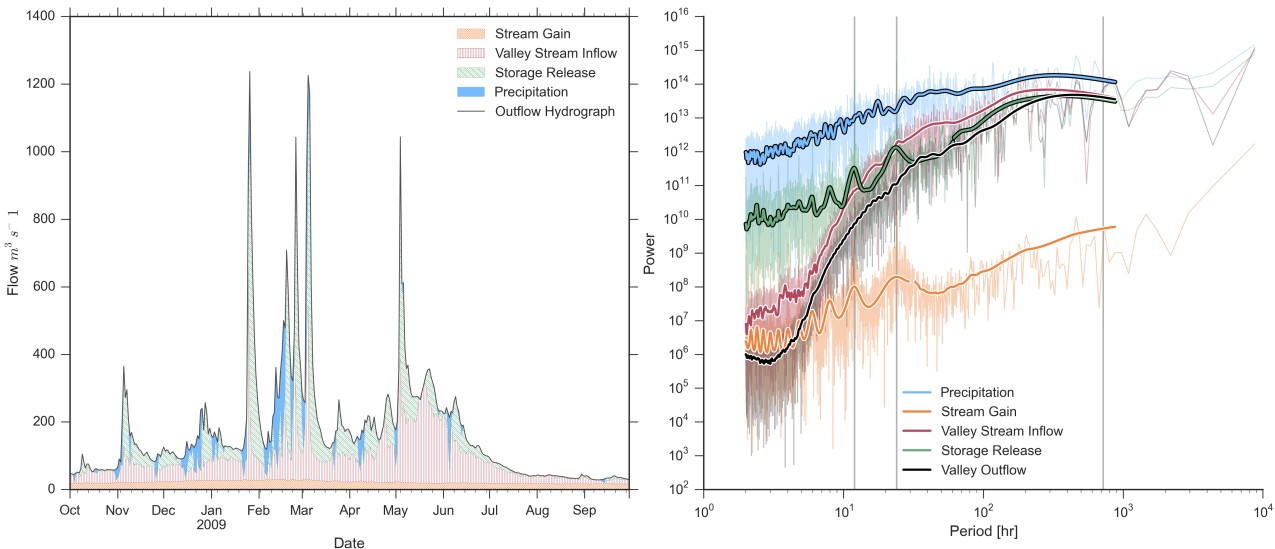

**Figure 10: (left) Average daily simulated San Joaquin River hydrograph separated into surface water budget components. (right) Periodogram showing the spectral properties of the hydrograph components. From left to right, vertical gray lines mark 12-hour, 24-hour, and 30-day periods.**

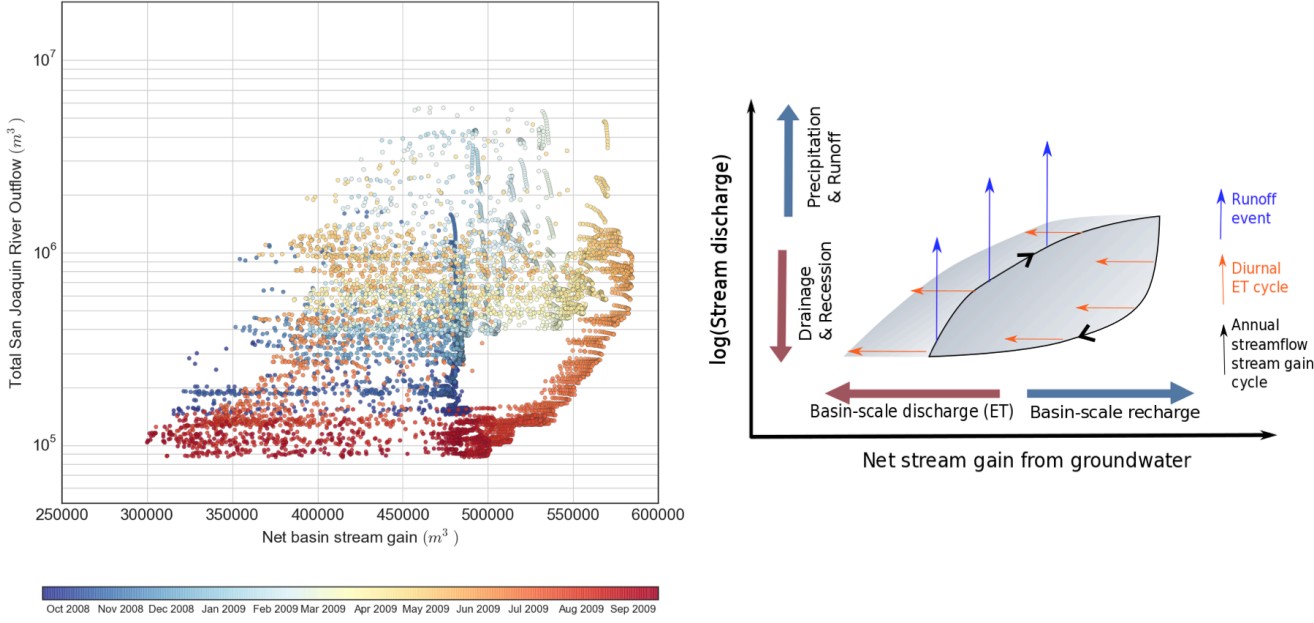

**Figure 11: Annual hysteresis in stream gain-streamflow relationship for the San Joaquin River. (left) Hourly simulated San Joaquin River flow is shown as a function of net gain from groundwater over the entire watershed. (right) Conceptual diagram highlighting the contributing processes to the annual hysteretic cycle.**

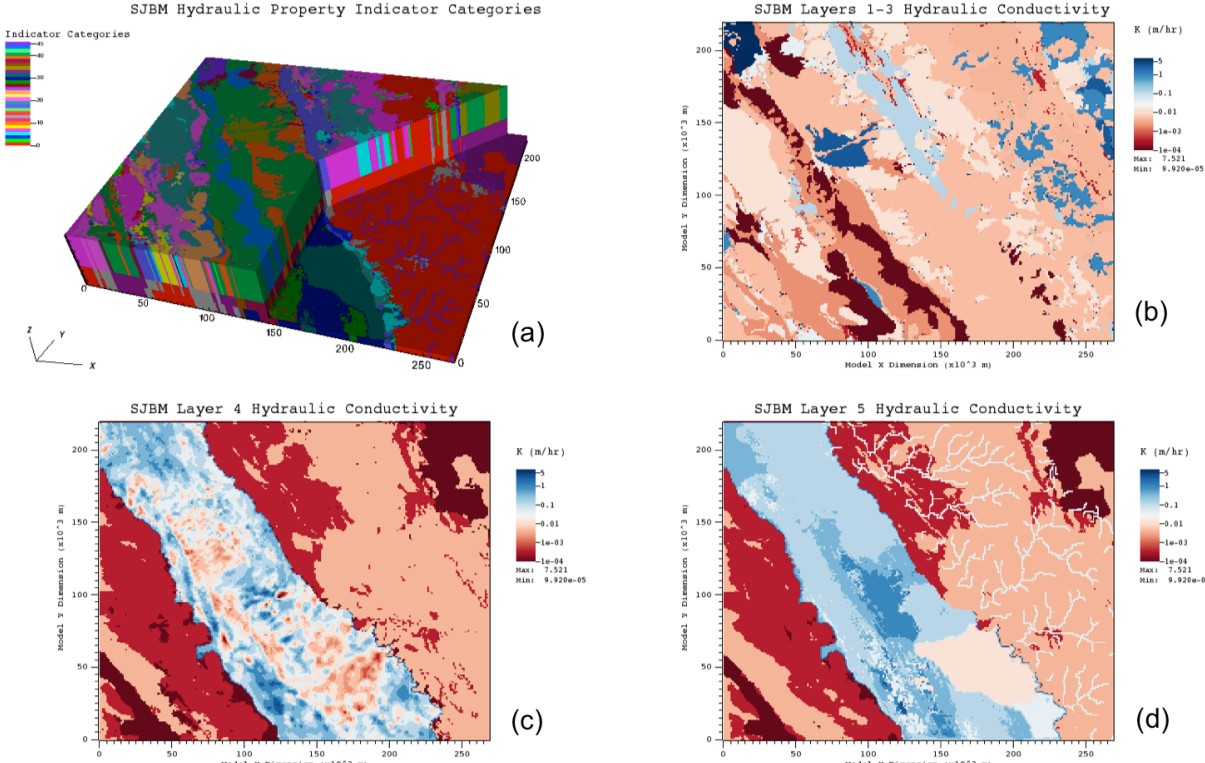

**Figure B1: Spatial distribution of subsurface hydraulic property indicator categories (a), hydraulic conductivity (m h⁻¹) in the top one meter of the model grid (layers 1-3) (b), hydraulic conductivity in model layer 4 (c), and hydraulic conductivity in the bottom model layer (layer 5) (d). Note in (a) that each layer is shown with an equal thickness and a vertical exaggeration of 100.**

