# Peer review of "Examining regional groundwater-surface water dynamics using an integrated hydrologic model of the San Joaquin River basin"

_Hydrology and Earth System Sciences, 2016_

## Referee Comment (RC1) · Anonymous Referee #1 · 10 Nov 2016

Gilbert and Maxwell provide an interesting study on a regional hydrologic model over large parts of California from groundwater to land surface. New class of models is applied trying to close the simulated hydrologic cylce in an integrated way. This is technically very advanced. Because the model does not include human water use, the results are interpreted as pre-development. Thus, they are difficult to verify with observations. Nevertheless the authors make an attempt to check plausibility. In ensuing steps, analyses (water budgets, stream-aquifer interactions, etc.) are performed in order to extract system responses that are to a large degree basin specific but also general. It is this generality that authors should highlight more to let reader know, which results are transferable.

The authors elaborate in detail the limitations in the comparison to observations, which have been impacted by water management practices and the lack of predevelopment gauge data as well as the influence of error prone atmospheric forcing. This is important. On the other hand, river discharge is not simulated explicitly as I understand, because no channel parameterization was used and the grid cells are 1km wide. Please discuss the potential impacts of this approximation on the comparison to available measurements in the context of infiltration and groundwater-surface interactions.

General comments

8, 1-10: Not sure if I understand. Why was snow accumulation/melt not taken into account. I was under impression that CLM-PF accounts for these processes (also section 4.3). 8, 13: Is there a reference for dry bias? 8, 32: I do not understand. Which NLDAS shortcoming exactly? 9, 6: data-drive? 9, 12: Could it also be that the model overestimates ET along river corridors, because of relatively coarse grid resolution? At 1km, river corridors can still not be resolved adequately. 10, 7-8: Seasonal variations recorded by GRACE are also influenced by anthropogenic impacts. Thus, GRACE and SJBM should not agree. 10, 20-30: Does this mean in turn that human impact cannot be determined from (GRACE) measurements since it is on the order of the error/uncertainty? 11, 12: Why is the unsaturated zone neglected? A figure might be useful. Figure 7: Plot change in storage on secondary axis. 12, 8-9: Remove, speculation. 12, 15.20: Please provide recharge estimates from other studies in the region. 13, 10-20: This is a useful analysis. What about structured heterogeneity in the aquifer. Could that also influence mountain block recharge? 16, 8-28: The interpretation of power spectra is overzealous. As authors pointed out, only one year of data are available. 16, 32: If the unsaturated zone serves as a filter between the saturated and unsaturated zone, why wasn't it included in the analysis?

The figures are of high quality. Please double check for typos.

---

## Referee Comment (RC2) · Anonymous Referee #2 · 25 Nov 2016

General comments

In this manuscript the authors aim to examine regional groundwater-surface water interactions and dynamics using an integrated model for the San Joaquin River basin. The authors start with a brief description of the model construction, followed by the validation of the model for groundwater levels, runoff and streamflow, snowpack, evapotranspiration and water storage. Following the validation, which the authors consider as satisfactory, the model is used for studying groundwater and land surface budgets, as well as groundwater-stream interactions. This is a high quality manuscript where the authors successfully employ a large number of data to validate a highly complex model, which then allows the authors to address highly relevant scientific questions

through the analysis of simulation results. Some of the choices during model design require a more detailed justification, such as the choice of a predevelopment situation or the selection of the year of interest (2009), please see specific comments below.

Specific comments

The paper is quite large and detailed, and yet lacks some detail on study area characterization, model calibration, sensitivity analysis or justification of certain choices.

In section 2 when the authors describe the conceptual model, they could give a first broad characterization of average annual precipitation (rainfall and snow), evapotranspiration. Also please indicate the total area of the basin in this section

Pg. 3. When the authors mention the main outflows, they do not consider groundwater. Do the authors think there might be any lateral groundwater outflows from the basin, through deep circulation and/or along faults.

Pg. 4. What is the estimated maximum thickness of the Corcoran Clays and to what extent may it hamper the upward flow of groundwater from deeper layers in the groundwater discharge area? As a consequence, what could be the impact of the oversimplification of the bottom layer? Was the presence of the Corcoran Clays taken into account when assigning the hydraulic conductivity values to the layer? Did the authors consider assigning a separate model layer to the clays? In Appendix B it is mentioned that they consider the current configuration as "a reasonable first approximation", does this mean that research on increasing vertical discretization of the aquifer is ongoing?

Pg. 4 The authors mention the Coast Range mountain blocks are built up of marine sedimentary rocks, whereas the Sierra mountain blocks are predominantly granite. The authors then conceptualize them as one and the same system with non-zero permeability through a depth of 500 m. Later on the authors perform a very interesting sensitivity analysis on mountain block permeability. My question is whether the authors initially thought of considering the Coast Range and Sierra mountain blocks as individual systems, since secondary porosity and permeability in carbonate rocks can be much higher due to dissolution/karstification phenomena, quite different from intrusive rocks.

I would like to read a bit more about the authors' justification for a using a (quasi-)predevelopment state to conceptualize the system in the model. I understand modelling a heavily altered state is much more complex. Notwithstanding, as the authors rightly mention, the calibration and validation of such a model is difficult. But more importantly, what can be said about the impact of human activities ("groundwater pumping, stream impoundments and reservoirs, or surface water diversions on the system") on the system, i.e. to what degree have they altered the dynamics that occurred in the predevelopment phase and are now simulated by the model? That is an important open question that remains to be answered (as the authors acknowledge in the summary and conclusions).

During parameterization of the model, did the authors perform any uncertainty/sensitivity analysis, besides the one mentioned for mountain block hydraulic conductivity? In particular, the authors opted for a single specific storage value for the entire model domain, and I would like them to explain how they calibrated this parameter and how sensitive the temporal groundwater storage changes are to this parameter. And regarding the K values, how do they vary spatially over the model domain?

For what reason did the authors use 2009 as your period of interest for simulation, and how does that year compare to an average year in terms of precipitation and temperature?

Comparison to observations: the authors mention that "considering aggregate behavior at a regional scale (1000-10000 km2) reduces some of the impact of local hydrologic perturbations sufficient to permit reasonable comparison of simulated and observed variables". On what do the authors base this assumption? And what about the regional hydrologic perturbations? It is known that the heavy groundwater pumping has caused

large regional perturbations, including water table lowering and land subsidence. In the water table map (section 4.1) the authors mention that land subsidence could explain part of the (high) discrepancies found between observed and modelled water table heights. Sparse point measurements could be another factor of influence, particularly close to the rivers. It would be good to see the location of the observation points in the map of Figure 2. The huge modelled hydraulic gradients on the western border of the area seem rather odd, in some cases decreasing from 90 m to 0 m in a few km. How can this be explained?

For runoff and streamflow (section 4.2) the authors use monthly "full natural flow" values provided by CA-DWR. How can these data be used to evaluate the correct representation of peak flows? In general, the authors acknowledge that for several reasons comparisons are difficult to make. They mention that the model tends to under-predict monthly flow volumes. What water balance parameter is then overestimated by the model? Strangely, in Fig. 3 I notice a systematic overestimation by the model of streamflow in the first months of the hydrological year. Please comment.

On evapotranspiration (section 4.3) how reliable are the results of the MOD16 global ET product? Have they been compared to other estimates of ET? I am also asking because I find the total recharge of the aquifer to be very low (Table 1, 2-3% of precipitation). For a sedimentary aquifer receiving diffuse recharge from direct rainfall, as well as significant mountain front recharge from rivers flowing out of the mountains onto the coarse-grained alluvial cones, I would expect recharge fractions of over 10% of watershed precipitation.

Regarding terrestrial water storage it is interesting that the authors use GRACE to verify the model results. One question I do have is how valid these comparisons are if you consider that large amounts of water are currently withdrawn for irrigation, thus constituting a significant loss factor that is not taken into account in the model but will show up in the GRACE signal. This is acknowledged by the authors, but they mention that the redistribution of water across the domain will have compensated these losses.

Please elaborate on this a bit.

On the groundwater budget (section 5.1) why is there a small aquifer-wide drop in the water table if the year is close to average in terms of climate and there are no groundwater abstractions?

Sections 5.1-5.3 are overall very well written and interesting to read. The figures used are clear, illustrative and well thought through.

Technical corrections

Pg. 2 ln 25: components Pg. 3 ln 10: temporal dynamics of

---

## Author Response (AR1)

Dr. Micha Werner
Editor, Hydrology and Earth System Sciences

Dear Dr. Werner,

We respectfully resubmit the enclosed manuscript titled "Examining regional groundwater-surface water dynamics using an integrated hydrologic model of the San Joaquin River basin" for review in Hydrology and Earth System Sciences journal. We thank the referees for their detailed reviews, thoughtful questions, and helpful suggestions. The revised manuscript has been improved in response to the discussion they prompted. In the attached documents we have provided responses to referee comments and have noted where the manuscript has been revised.

We thank you as well for your review of the manuscript and discussion. In revising the manuscript we sought to clarify the points indicated by both referees and have included a revised Figure 7 and a new Figure B1 to better communicate model results and inputs, respectively.

I confirm that this manuscript contains only original data that has not been published in another journal and, if accepted, will not be published elsewhere without written consent. Thank you for your time and I look forward to hearing from you regarding the status of this submission.

Sincerely,

James M. Gilbert
(now at the US Bureau of Reclamation)
jmgilbert@usbr.gov

**Paper Title:** Examining regional groundwater-surface water dynamics using an integrated hydrologic model of the San Joaquin River basin

**Paper #:** HESS-2016-488

Responses to Reviewers
- *Comments from the referees are shown in italics*
- Author responses are shown in plain text
- **Notes indicating changes to the manuscript in response to referee comments are shown in bold**

Anonymous Referee #1

*Comment 1: Authors should better distinguish and emphasize the findings from analysis of San Joaquin basin system responses that are transferable or generalizable beyond this specific basin.*

This is a good point. We will revise the manuscript to better emphasize the more generalizable findings.

**\*Text was added to page 20, new lines 16-18 to note the potential generalizability beyond the San Joaquin of the hydrograph separation and hysteretic stream-groundwater behavior based on basin properties**

*Comment 2: "river discharge is not simulated explicitly as I understand, because no channel parameterization was used and the grid cells are 1 km wide. Please discuss the potential impacts of this approximation on the comparison to measurements in the context of infiltration and groundwater-surface water interactions."*

River discharge is explicitly simulated by ParFlow inasmuch as all ponded water at the land surface is subject to the same mathematical representation of the runoff process, i.e. the kinematic approximation of the shallow water equations, that results in that water being routed downstream. Overland flow water that concentrates in zones of converging slopes is generally coincident with the location of mapped stream or river channels but, as correctly noted by the referee, there is no separate parameterization of flow for *a priori* designated river channels. The ParFlow code conserves mass, so, although the routed ponded water is spread out over a 1 km-wide cell, river discharge at a point is an accessible model output through a calculation given the simulated ponded depth, slope, and roughness coefficient.

No modeling approach can represent a system with perfect, complete fidelity. The approach used in this study ensures that the overland flow system and subsurface flow system are solved in a universally consistent, implicit manner at each time step. This provides the benefit of capturing complex combinations of subsurface and surface-driven overland flow generation mechanisms but may limit representation of other aspects of the surface water system – like channel flow. We are not aware of any studies that rigorously examine the impacts of grid

resolution for complex processes like groundwater-surface water interaction and infiltration for a complex, real-world basin using a model like ParFlow. Thus, we compare to observations in this study to help identify how well a basin-scale system can be represented with 1-km resolution. While some tests on idealized hillslopes suggest increasing grid size in ParFlow tends to reduce and delay peak saturation-excess type runoff over short distances (Maxwell, 2013), it is not necessarily straightforward to extrapolate these phenomena to larger, more complex, real world watershed simulations. The referee raises an important question, one that we believe warrants further investigation to better guide future modeling efforts.

**\*New text was added to page 5, lines 13-14, to better clarify that ParFlow does not include a river parameterization but does simulate river discharge.**

*8, 1-10: Regarding runoff ratio comparisons: "Why was snow accumulation/melt not taken into account?"*

ParFlow-CLM does simulate snow accumulation and melt. The simulated runoff is lower than would exist if the winter precipitation fell as rain rather than snow, just as spring/summer simulated runoff is higher than it would be for the given precipitation amount because of snowmelt contributions. These processes are also embedded in the streamflow response measured (and adjusted) in the naturalized flow values that form the basis of the comparison. Comparing a runoff ratio that converts snow accumulation into an equivalent melt volume (and then reducing subsequent runoff increases by the corresponding snowmelt amount) would require extra assumptions and calculations in both the simulated and observed components. These extra assumptions would obscure and complicate the comparison between model and observations.

**\*Additional text was included on page 8, new lines 26-27 to better indicate that monthly runoff ratios reflect snow accumulation and melt rather than trying to convert snowpack into equivalent runoff.**

*8, 13: Is there a reference for dry bias?*
*8,32: Which NLDAS shortcoming?*

We note on page 9, lines 1-3 that several studies have demonstrated the tendency for atmospheric forcing products to under-resolve orographic effects when the product resolution is lower than the terrain of interest. Additionally, comparisons to meteorological station data during the development of this model suggested a tendency toward under-representing precipitation at mid- and high elevations across the Sierra Nevada in the model. We will revise to make this point clearer and more specific.

**\*New text was added to the end of Section 4.2, page 8 new lines 33-34 and page 9, new lines 1-3.**

*9, 12: Could it be that the model overestimates ET along river corridors, because of relatively coarse grid resolution?*

Yes, this is another potential explanation that bears mentioning. However, we expect the contribution of this effect to be mitigated somewhat by the fact that a saturated 1-km cell permits full potential transpiration to occur. The near-surface riparian-zone water table of the unimpacted system likely would have supported near-potential transpiration for much of the year, thus the key discrepancy is more likely bare soil evaporation. In cells that correspond to river channels that are wetted in both the predevelopment and modern systems, the ParFlow-CLM model could be overestimating evaporation and, to an extent that depends on the differences in the riparian water table configuration, transpiration.

**\*A sentence was added (page 10, new line 5-6) to include this a potential explanation**

*10, 7-8: Season variations recorded by GRACE are also influenced by anthropogenic impacts. Thus GRACE and SJBM should not agree.*
*10, 20-30: Does this mean in turn that human impact cannot be determined from (GRACE) measurements since it is on the order of the error/uncertainty?*

The comparison to GRACE perhaps should have received a more complete discussion in the original manuscript. We intended the comparison of the SJBM results to GRACE to be a general check on the ability of the model to simulate an annual cycle of terrestrial water storage change. We acknowledge that the comparison is complicated by the fact that the SJBM does not include anthropogenic (irrigation, reservoir & canal operations, etc), but given that 1) these activities are interconnected and potentially offsetting (e.g. imports of water may balance some of the increase in ET associated with groundwater extraction for irrigation); 2) studies that attempt to assign the groundwater signal to a portion of the overall GRACE signal do not incorporate the offsetting impact of applying extracted water to the land surface in the land surface models used in the calculations (Famiglietti et al., 2011; Scanlon et al., 2012); and 3) native vegetation has some non-zero ET such that ET of irrigation water cannot be considered a wholesale increase in loss from the system compared to the SJBM-modeled "natural" state, the impact of water management and irrigation on the GRACE signal is far from settled.

Disentangling the effects of San Joaquin Basin surface water imports and exports, consumptive use losses, and seepage/recharge losses associated with (local and imported) irrigation water is difficult and would best be attempted through a model that could accommodate both the natural and anthropogenic processes at play. The SJBM model presented here provides a baseline "natural" state that could complement such an analysis but that work is beyond the scope of this study. Without such a model, coarse estimates of the impact of human activity on terrestrial water storage can be derived from a naïve assessment of basin water balance values such as those published by the California Department of Water Resources in their 2013 California Water Plan Update (Bulletin 160-13, California Department of Water Resources, 2014).

From Table SJR-19 in the San Joaquin Basin Report in Volume 2, we can estimate that the net of imports and exports from/to the San Joaquin River basin (area 15214 $mi^2$ or 39404 $km^2$) for water year 2009 is -1.49 x $10^6$ acre-feet, or, converted to an equivalent depth over the basin area, -3.125 cm. Assuming that the ratio of consumptive use to applied water represented by values in Table SJR-19 holds for groundwater, and that groundwater extraction and application is fairly estimated at 3.5 x $10^6$ acre-feet (~38% of applied water, from Table SJR-16), or 7.34 cm over the basin, the net loss of groundwater is approximately 4.88 cm (consumptive use/applied water = 0.66 * Applied GW = consumptively used GW). The combined effect of the net export of water and groundwater consumptive use, based on this naïve approach, is then a net loss of 8.0 cm. If we assume this 8 cm manifests as an absolute increase in seasonal terrestrial water storage amplitude and could be added directly to the SJBM terrestrial water storage, the SJBM result would still fall nearly within the error bands of the GRACE signal.

We readily agree that important uncertainties exist in both the GRACE product and our ability to connect it with a quantitative depiction of terrestrial hydrologic components through *in situ* observations and models. Many models and observations have been compared to GRACE data products and have helped improve our understanding of both model and data product strengths and weaknesses. We extend this trend by comparing a basin-scale integrated hydrologic model to GRACE data, the first such comparison that we are aware of, and consider this an important step in the advancement of our use of both tools. Furthermore, the questions raised in this discussion are important ones that should motivate focused research on the topic.

**\*This point was summarized and added to hopefully clarify the GRACE-model comparison presented in the manuscript. New text was added from page 11, new line 22 to page 12, new line 6.**

*11, 12: Why is the unsaturated zone neglected? A figure might be useful*
*16, 32: If the unsaturated zone serves as a filter between the saturated and unsaturated zone, why wasn't it included in the analysis?*

We are unsure precisely what the referee means with these comments. ParFlow simulates variably saturated flow, so the unsaturated zone is inherently a part of the analysis. We were interested in examining traditionally separated hydrologic components (i.e. groundwater, surface water) as they evolve in a fully integrated model at a basin scale. Although we could explicitly include the unsaturated zone in the analysis, the effect is implicit on both the saturated and surface systems as we have noted in the manuscript. If the referee is referring to a different representation or treatment of the unsaturated zone than we have presented, we would be interested to consider it to as part of future work.

**\*Text was added to page 12, new lines 19-22 to clarify the implicit inclusion of the unsaturated zone in the analysis.**

*12, 15-20: Provide recharge estimates from the studies in the region*

Such values would be good for reference. We will add them to the revised manuscript.

**\*Text that provides recharge values from the two key regional groundwater modeling studies of the Central Valley has been included on page 13, new lines 15-18.**

*13, 10-20:* Regarding analysis of mountain block recharge: *What about structured heterogeneity in the aquifer? Could that also influence mountain block recharge?*

This is a good point. We would expect structured aquifer heterogeneity to influence local mountain block recharge but extrapolating to an aggregate aquifer-scale impact is difficult to project. This would be an interesting question to test as it would highlight whether the mountain-valley gradient (topographic control) is a more important factor than local geologic control.

*16, 8-28: The interpretation of power spectra is overzealous.*

We are unsure what portion of our interpretation overreaches. We intended the analysis to provide some insight for higher frequency (diurnal to multi-day) temporal scaling of hydrologic components. We believe the one year of hourly data used in the study is sufficient to support this – i.e. we don't expect that extending the time series through a multi-year simulation would substantively change the spectra at high frequencies.

*Typographical errors as noted by referee will be corrected/adjusted:*
Page 9, line 6: 'data-drive' will be corrected to 'data-driven'
Figure 7: Plot change in storage on secondary axis
Page 12, line 8-9: remove "speculation"

**The Figure 7 in the original manuscript had inadvertently used hourly storage change time series with daily groundwater fluxes. We thank the referee for pointing this out and have now corrected the plot so that variability across each groundwater flux type and storage change are more visible on the same scale.**

Specific comments:

*Section 2: "…give a first broad characterization of average annual precipitation (rainfall and snow), evapotranspiration. Also please indicate the total area of the basin in this section"*

We agree this would be helpful background information and will incorporate this into a revised system description.

**The total area of the basin has been added to Page 3, new line 15.**
**Text describing the average annual precipitation in the Sierra Nevada and Central Valley portions of the basin along with average ET and streamflow outflows has been added to Page 3, new lines 25-27 and Page 3, new lines 30-31, respectively.**

*Page 3. When the authors mention the main outflows, they do not consider groundwater. Do the authors think there might be any lateral groundwater outflows from the basin, through deep circulation and/or along faults.*

The use of no-flow boundaries in the subsurface portion of the model is a simplification – we would expect some non-zero groundwater flux across those boundaries in real life. Similarly, some amount of water is also likely flowing below the 500-m depth that we simulate. We consider these groundwater fluxes negligble for the simulated system of interest based on the following: 1) the approximated predevelopment water table of (Williamson et al., 1989) shows a relatively flat water table in the region coinciding with the southern model boundary in the Central Valley; 2) the east and west model boundaries are beyond the topographic San Joaquin Basin delineation and are characterized by outflowing surface water systems – the absence of groundwater flow along these boundaries does not likely match reality but are sufficiently distant from our area of interest for the error to be minimal; 3) Faunt (2009) shows that groundwater flow to the Delta (the northern SJBM model boundary in the Central Valley) as a fraction of the overall groundwater budget of the San Joaquin River basin is comparatively small.

**Text describing the possibility of groundwater outflow and justification for being considered minimal has been added to Page 4, new lines 1-5.**

*Page 4. What is the estimated maximum thickness of the Corcoran Clays and to what extent may it hamper the upward flow of groundwater from deeper layers in the groundwater discharge area? As a consequence, what could be the impact of the oversimplification of the bottom layer? Was the presence of the Corcoran Clays taken into account when assigning the hydraulic conductivity values to the layer? Did the authors consider assigning a separate model layer to the clays? In Appendix B it is mentioned that they consider the current configuration as "a reasonable first approximation", does this mean that research on increasing vertical discretization of the aquifer is ongoing?*

Previous studies report an approximate maximum thickness of the Corcoran Clay to be 61 m (200ft) (Davis et al., 1959; Faunt, 2009; Page, 1986). This clay, where contiguous, acted as a confining layer in the unimpacted predevelopment system. The Corcoran clay is not explicitly represented in the subsurface properties used in the SJBM so vertical pressure (head) and associated gradients in the lower portion of the modeled aquifer would tend to be lower than in the real predevelopment system. The SJBM more directly approximates the upper, semi-confined to unconfined aquifer system in the vicinity of the Corcoran clay. Furthermore, a large component of the flow in the Central Valley aquifer system occurs laterally, though, and this component is well-represented by the model configuration. The hydrostratigraphy in the SJBM is admittedly simple and limits the ability to simulate vertical gradients necessary for replicating multi-level groundwater extraction and drawdowns. Work to improve the vertical resolution of the model is ongoing.

**Text clarifying the absence of an explicit representation of the Corcoran Clay has been added to Appendix A, page 21, new lines 11-14.**

*Pg. 4 The authors mention the Coast Range mountain blocks are built up of marine sedimentary rocks, whereas the Sierra mountain blocks are predominantly granite. The authors then conceptualize them as one and the same system with non-zero permeability through a depth of 500 m. Later on the authors perform a very interesting sensitivity analysis on mountain block permeability. My question is whether the authors initially thought of considering the Coast Range and Sierra mountain blocks as individual systems, since secondary porosity and permeability in carbonate rocks can be much higher due to dissolution/karstification phenomena, quite different from intrusive rocks.*

The Coast Range and Sierra Nevada are represented in the model by different hydraulic properties in an attempt to capture the differences due to the geologic factors the referee mentions. We describe the mountain blocks as being simulated as part of the same system because that portion of the hydrologic system is subject to the same governing physical processes in the simulation as the Central Valley – in contrast to the commonly employed approach that includes the mountain blocks only as a predefined boundary condition.

**Text clarifying that the Coast Range and Sierra Nevada are represented in the model by different hydraulic properties was added to Page 4, new lines 15-16.**

*I would like to read a bit more about the authors' justification for a using a (quasi-)predevelopment state to conceptualize the system in the model. I understand modelling a heavily altered state is much more complex. Notwithstanding, as the authors rightly mention, the calibration and validation of such a model is difficult. But more importantly, what can be said about the impact of human activities ("groundwater pumping, stream impoundments and reservoirs, or surface water diversions on the system") on the system, i.e. to what degree have they altered the dynamics that occurred in the predevelopment phase and are now simulated by*

*the model? That is an important open question that remains to be answered (as the authors acknowledge in the summary and conclusions).*

As the referee points out, understanding the full impact of human-driven hydrologic change such as what results from the history of water management infrastructure and groundwater extraction in the San Joaquin River basin region, requires comparison to some baseline or unimpacted benchmark. Where the history of change is recent, this baseline condition may be determined through the measurement record. In the Central Valley, water extraction and management activities predate much of the measurement record. This motivates the use of a model to estimate and constrain aspects of the predevelopment system in general and the development of the SJBM specifically.

**Text was added clarifying the motivation for using a predevelopment conceptualization of the system as the basis for the model, page 4, new lines 27-31.**

*During parameterization of the model, did the authors perform any uncertainty/sensitivity analysis, besides the one mentioned for mountain block hydraulic conductivity? In particular, the authors opted for a single specific storage value for the entire model domain, and I would like them to explain how they calibrated this parameter and how sensitive the temporal groundwater storage changes are to this parameter. And regarding the K values, how do they vary spatially over the model domain?*

A rigorous sensitivity analysis of the spatially distributed hydraulic parameters or land surface properties was not performed during model development due to simulation time and file storage constraints. Specific storage was set at a constant reference value based on previous studies (e.g. Maxwell et al, 2015) and anecdotal evidence that overall storage dynamics for a basin-scale simulation without dynamic groundwater extraction are relatively insensitive to specific storage. By definition, the compressible storage is orders of magnitude lower than incompressible storage. We expect the role of specific storage to be greater for a dynamically impacted system and intend to refine specific storage representation in the model as part of ongoing work to improve resolution of the model.

Hydraulic conductivity values vary throughout the domain based on datasets developed through previous studies (e.g. Faunt, 2009). We can provide figures in the appendices to show this distribution.

**\*Figure B1 was added to Appendix B to illustrate the three-dimensional spatial variability of subsurface properties as assigned to the San Joaquin Basin Model domain.**

*For what reason did the authors use 2009 as your period of interest for simulation, and how does that year compare to an average year in terms of precipitation and temperature?*

We chose water year 2009 for several reasons. First, we wanted to simulate a recent year to maximize the availability and coverage of *in situ* and remote sensing-derived data for

comparison to model results. Second, we wanted to simulate an approximately average, in a climatological sense, water year to ensure that simulation results would reflect the response of the hydrologic system rather than being dominated by a particular year's weather (e.g. extreme drought, flood, etc). It is difficult to find a perfectly "average" year for every meterological variable, but we chose 2009 based on its nearly average precipitation and temperature conditions. For the San Joaquin river basin for water year 2009: 1) average annual temperature is slightly warmer than the historical average (1 degree C compared to 1895-1970); 2) annual precipitation is slightly drier than the historical average (40.2 cm in 2009; 50.1 cm the average for 1895-1980) (NOAA Climate Data at Glance: https://www.ncdc.noaa.gov/cag/time-series/us/4/5/); and 3) state of California basin water supply index for the San Joaquin River basin is slightly drier than normal.

**\*Text was added to page 6, new lines 10-14, to describe why 2009 was chosen and its relative precip/temperature compared to historical averages.**

*Comparison to observations: the authors mention that "considering aggregate behavior at a regional scale (1000-10000 km2) reduces some of the impact of local hydrologic perturbations sufficient to permit reasonable comparison of simulated and observed variables". On what do the authors base this assumption? And what about the regional hydrologic perturbations? It is known that the heavy groundwater pumping has caused large regional perturbations, including water table lowering and land subsidence.*

The vast majority of the anthropogenic water use occurs in the Central Valley while the SJBM extends into the Sierra Nevada. Considering the basin as a whole, the water budget is dominated by fluxes into and out of the Sierra Nevada such that a large portion of variation in total volume of water in the system depends on that region. We acknowledge that a mismatch in absolute volumes of water results from historical impacts to the Central Valley system, but the seasonal pattern of variation that results from natural (meteorological) forcing should remain fairly similar for both the natural and modern (impacted) systems.

**\*Text clarifying this point has been added to page 6, new lines 25-26.**

*In the water table map (section 4.1) the authors mention that land subsidence could explain part of the (high) discrepancies found between observed and modelled water table heights. Sparse point measurements could be another factor of influence, particularly close to the rivers. It would be good to see the location of the observation points in the map of Figure 2. The huge modelled hydraulic gradients on the western border of the area seem rather odd, in some cases decreasing from 90 m to 0 m in a few km. How can this be explained?*

The observations that supported the water table contour map from Williamson et al (1989) were not immediately available but are undoubtedly sparse, especially near rivers as the referee notes. Large gradients at the Central Valley edge result from high topographic gradients combined with, in locations, abrupt changes in hydraulic conductivity. We believe this to be the case with the high gradients noted along the western border of the Valley.

*For runoff and streamflow (section 4.2) the authors use monthly "full natural flow" values provided by CA-DWR. How can these data be used to evaluate the correct representation of peak flows? In general, the authors acknowledge that for several reasons comparisons are difficult to make. They mention that the model tends to under-predict monthly flow volumes. What water balance parameter is then overestimated by the model? Strangely, in Fig. 3 I notice a systematic overestimation by the model of streamflow in the first months of the hydrological year. Please comment.*

These naturalized flow data provide a means to assess seasonal peak flows – i.e. the flows that result from annual snowmelt and spring rains in the region. They do not provide event-scale information however.

The model tends to underpredict monthly flows, especially in the late spring and early summer period. We believe this to be the result of 1) too little precipitation in the meteorological inputs and 2) a very modest bias toward partitioning water to groundwater instead of runoff. This second factor shows up as a slightly higher runoff in the September-October time frame as noted.

*On evapotranspiration (section 4.3) how reliable are the results of the MOD16 global ET product? Have they been compared to other estimates of ET? I am also asking because I find the total recharge of the aquifer to be very low (Table 1, 2-3% of precipitation). For a sedimentary aquifer receiving diffuse recharge from direct rainfall, as well as significant mountain front recharge from rivers flowing out of the mountains onto the coarse-grained alluvial cones, I would expect recharge fractions of over 10% of watershed precipitation.*

The low recharge fraction is a result of the fact that the watershed precipitation amount used in the calculation includes high Sierra Nevada mountain precipitation (where most of the watershed precipitation falls) while the recharge considered is limited to the Central Valley portion of the model domain (where relatively little precipitation falls). This recharge fraction value shows the relative partitioning of precipitation in the mountains between runoff (that can potentially recharge at the mountain front) and groundwater (that contributes to mountain block recharge).

We acknowledge that no measurement, remotely sensed data, or, in the case of the MOD16 ET product, algorithmically-derived data product, is perfect. The MOD16 ET data product was deemed sufficiently accurate based on observation validations published by (Mu et al., 2011).

**\*A sentence clarifying that the recharge fraction arises from total watershed precipitation rather than the precipitation amount that falls on the Central Valley itself was added to page 14, new lines 25-27.**

*Regarding terrestrial water storage it is interesting that the authors use GRACE to verify the model results. One question I do have is how valid these comparisons are if you consider that*

*large amounts of water are currently withdrawn for irrigation, thus constituting a significant loss factor that is not taken into account in the model but will show up in the GRACE signal. This is acknowledged by the authors, but they mention that the redistribution of water across the domain will have compensated these losses. Please elaborate on this a bit.*

See discussion in response to the same question from Referee #1.

*On the groundwater budget (section 5.1) why is there a small aquifer-wide drop in the water table if the year is close to average in terms of climate and there are no groundwater abstractions?*

This small drop in aquifer storage occurs because water year 2009 is slightly drier than average and the model, despite a long period of dynamic spin-up, still loses slightly more water than it gains. This small drop was considered to be acceptable in the context of larger water balance components and is consistent with the GRACE trend for the region.

**\*A sentence has been added to note and explain this small drop in aquifer storage, page 15, new lines 10-11.**

*Sections 5.1-5.3 are overall very well written and interesting to read. The figures used are clear, illustrative and well thought through*.

*Technical corrections:*
 *Pg. 2 ln 25: components*
*Pg. 3 ln 10: temporal dynamics of*

California Department of Water Resources: California Water Plan Update 2013, Bulletin, California Department of Water Resources., 2014.

Davis, G. H., Green, J. H., Olmsted, F. H. and Brown, D. W.: Ground-water conditions and storage capacity in the San Joaquin Valley, California, Report. [online] Available from: http://pubs.er.usgs.gov/publication/wsp1469, 1959.

Famiglietti, J. S., Lo, M., Ho, S. L., Bethune, J., Anderson, K. J., Syed, T. H., Swenson, S. C., de Linage, C. R. and Rodell, M.: Satellites measure recent rates of groundwater depletion in California's Central Valley, Geophys. Res. Lett., 38(3), L03403, doi:10.1029/2010GL046442, 2011.

Faunt, C. C., Ed.: Groundwater Availability of the Central Valley Aquifer, California, United States Geological Survey. [online] Available from: http://pubs.usgs.gov/pp/1766/, 2009.

Maxwell, R. M.: A terrain-following grid transform and preconditioner for parallel, large-scale, integrated hydrologic modeling, Adv. Water Resour., 53, 109–117, doi:10.1016/j.advwatres.2012.10.001, 2013.

Maxwell, R. M. and Condon, L. E. and Kollet, S. J.: A high-resolution simulation of groundwater and surface water over most of the continental US with the integrated hydrologic model ParFlow v3, Geoscientific Model Development, 8, 923-937, doi: 10.5194/gmd-8-923-2015

Mu, Q., Zhao, M. and Running, S. W.: Improvements to a MODIS global terrestrial evapotranspiration algorithm, Remote Sens. Environ., 115(8), 1781–1800, doi:10.1016/j.rse.2011.02.019, 2011.

Page, R. W.: Geology of the fresh ground-water basin of the Central Valley, California, with texture maps and sections, Report. [online] Available from: http://pubs.er.usgs.gov/publication/pp1401C, 1986.

Scanlon, B. R., Longuevergne, L. and Long, D.: Ground referencing GRACE satellite estimates of groundwater storage changes in the California Central Valley, USA, Water Resour. Res., 48(4), W04520, doi:10.1029/2011WR011312, 2012.

Williamson, A. K., Prudic, D. E. and Swain, L. A.: Ground-water flow in the Central Valley, California, Professional Paper, United States Geological Survey. [online] Available from: http://pubs.er.usgs.gov/publication/pp1401D (Accessed 10 November 2014), 1989.

**List of Relevant Changes:**

*Updated contact information for James Gilbert – he is now staff at the Bureau of Reclamation

*Text was added to page 20, new lines 16-18 to note the potential generalizability beyond the San Joaquin of the hydrograph separation and hysteretic stream-groundwater behavior based on basin properties

*New text was added to page 5, lines 13-14, to better clarify that ParFlow does not include a river parameterization but does simulate river discharge.

*Additional text was included on page 8, new lines 26-27 to better indicate that monthly runoff ratios reflect snow accumulation and melt rather than trying to convert snowpack into equivalent runoff.

*New text was added to the end of Section 4.2, page 8 new lines 33-34 and page 9, new lines 1-3 to better clarify NLDAS tendency for dry bias over more highly resolved complex terrain

*A sentence was added (page 10, new line 5-6) to include model resolution as another potential explanation for local ET mismatches

*Discussion on the effect of pumping/water management and GRACE data was added to hopefully clarify the GRACE-model comparison presented in the manuscript. New text was added from page 11, new line 22 to page 12, new line 6.

*Text was added to page 12, new lines 19-22 to clarify the implicit inclusion of the unsaturated zone in the analysis.

*Text that provides recharge values from the two key regional groundwater modeling studies of the Central Valley has been included on page 13, new lines 15-18.

*Figure 7 was modified with a version that includes consistent flux and storage change components, all visible on the same y-axis

*The total area of the basin has been added to Page 3, new line 15.
Text describing the average annual precipitation in the Sierra Nevada and Central Valley portions of the basin along with average ET and streamflow outflows has been added to Page 3, new lines 25-27 and Page 3, new lines 30-31, respectively.

*Text describing the possibility of groundwater outflow and justification for being considered minimal has been added to Page 4, new lines 1-5.

*Text clarifying the absence of an explicit representation of the Corcoran Clay has been added to Appendix A, page 21, new lines 11-14.

* Text clarifying that the Coast Range and Sierra Nevada are represented in the model by different hydraulic properties was added to Page 4, new lines 15-16.

* Text was added clarifying the motivation for using a predevelopment conceptualization of the system as the basis for the model, page 4, new lines 27-31.

* Text was added to page 6, new lines 10-14, to describe why 2009 was chosen and its relative precip/temperature compared to historical averages.

*Text the averaging or smoothing out of perturbations over the basin when considering more natural hydrologic cycles in the Sierra Nevada was added to page 6, new lines 25-26.

*A sentence clarifying that the recharge fraction arises from total watershed precipitation rather than the precipitation amount that falls on the Central Valley itself was added to page 14, new lines 25-27.

*A sentence has been added to note and explain this small drop in aquifer storage, page 15, new lines 10-11.

*Figure B1 was added to Appendix B to illustrate the three-dimensional spatial variability of subsurface properties as assigned to the San Joaquin Basin Model domain.

[revised manuscript text omitted]